# The RRM-mediated RNA binding activity in *T. brucei* RAP1 is essential for *VSG* monoallelic expression

Amit Kumar Gaurav [1,10,13], Marjia Afrin[1,11,13], Xian Yang[2,3,13], Arpita Saha[1,12], S. K. Abdus Sayeed[1], Xuehua Pan[2,3], Zeyang Ji[4], Kam-Bo Wong[5], Mingjie Zhang [4,6], Yanxiang Zhao [2,3] ✉ & Bibo Li [1,7,8,9] ✉

*Trypanosoma brucei* is a protozoan parasite that causes human trypanosomiasis. Its major surface antigen VSG is expressed from subtelomeric loci in a strictly monoallelic manner. We previously showed that the telomere protein *Tb*RAP1 binds dsDNA through its $_{737}$RKRRR$_{741}$ patch to silence *VSGs* globally. How *Tb*RAP1 permits expression of the single active *VSG* is unknown. Through NMR structural analysis, we unexpectedly identify an RNA Recognition Motif (RRM) in *Tb*RAP1, which is unprecedented for RAP1 homologs. Assisted by the $_{737}$RKRRR$_{741}$ patch, *Tb*RAP1 RRM recognizes consensus sequences of *VSG* 3'UTRs in vitro and binds the active *VSG* RNA in vivo. Mutating conserved RRM residues abolishes the RNA binding activity, significantly decreases the active *VSG* RNA level, and derepresses silent *VSGs*. The competition between *Tb*RAP1's RNA and dsDNA binding activities suggests a *VSG* monoallelic expression mechanism in which the active *VSG*'s abundant RNA antagonizes *Tb*RAP1's silencing effect, thereby sustaining its full-level expression.

Monoallelic gene expression (MAE), or allelic exclusion, is essential for many organisms. Notable examples include genome imprinting, X chromosome inactivation, and random monoallelic expression of autosomal genes in mammals[1]. Many genes that undergo MAE encode cell surface receptors. For example, each human and mouse olfactory sensory neuron expresses only one odorant receptor gene[1]. Several microbial pathogens, including *Trypanosoma brucei*, employ MAE as part of their antigenic variation strategy[2–5].

*Trypanosoma brucei* is a protozoan parasite that causes human African trypanosomiasis. It sequentially expresses distinct Variant

[1]Center for Gene Regulation in Health and Disease, Department of Biological, Geological, and Environmental Sciences, College of Arts and Sciences, Cleveland State University, 2121 Euclid Avenue, Cleveland, OH 44115, USA. [2]Department of Applied Biology and Chemical Technology, State Key Laboratory of Chemical Biology and Drug Discovery, The Hong Kong Polytechnic University, Hung Hom, Kowloon, Hong Kong, People's Republic of China. [3]The Hong Kong Polytechnic University Shenzhen Research Institute, Shenzhen 518057, People's Republic of China. [4]Division of Life Science, Hong Kong University of Science and Technology, Clear Water Bay, Kowloon, Hong Kong, People's Republic of China. [5]Centre for Protein Science and Crystallography, School of Life Sciences, State Key Laboratory of Agrobiotechnology, The Chinese University of Hong Kong (CUHK), Shatin, Hong Kong, China. [6]School of Life Sciences, Southern University of Science and Technology, Shenzhen 518055, People's Republic of China. [7]Case Comprehensive Cancer Center, Case Western Reserve University, 10900 Euclid Avenue, Cleveland, OH 44106, USA. [8]Department of Inflammation and Immunity, Lerner Research Institute, Cleveland Clinic, 9500 Euclid Avenue, Cleveland, OH 44195, USA. [9]Center for RNA Science and Therapeutics, Case Western Reserve University, 10900 Euclid Avenue, Cleveland, OH 44106, USA. [10]Present address: The Wistar Institute, Philadelphia, PA 19104, USA. [11]Present address: Institute for Stem cell Biology and Regenerative Medicine, Stanford School of medicine, Stanford University, Palo Alto, CA 94305, USA. [12]Present address: Telomeres and Telomerase Group, Molecular Oncology Program, Spanish National Cancer Centre (CNIO), Madrid 28029, Spain. [13]These authors contributed equally: Amit Kumar Gaurav, Marjia Afrin, Xian Yang. ✉e-mail: yanxiang.zhao@polyu.edu.hk; b.li37@csuohio.edu

Surface Glycoproteins (VSGs), its major surface antigen, to evade the host's immune surveillance. *VSG* is monoallelically transcribed by RNA polymerase I[6] from one of the ~15 nearly identical *VSG* expression sites (ESs)[7,8]. In these subtelomeric polycistronic transcription units, *VSG* is always the last gene located within 2 kb of the telomeric repeats[7,8]. Parasites that express multiple VSGs are more rapidly eliminated by the host[9], underscoring the importance of *VSG* MAE for *T. brucei* survival. Many factors affect *VSG* MAE, such as nuclear lamina, the inositol phosphate pathway[2,3], transcription elongation[10,11], and a subtelomere and *VSG*-associated VEX complex[12-14].

Our previous studies have demonstrated that *Tb*RAP1, a nuclear and essential telomere protein, is a key regulator of *VSG* MAE[15-18]. Depletion of *Tb*RAP1 leads to derepression of silent *VSGs* up to a thousand-fold[15-18]. The *Tb*RAP1-mediated silencing is stronger at loci closer to the telomere than those further away[15]. We recently reported that *Tb*RAP1 possesses a dsDNA binding activity mediated by its R/K patch ($_{737}$RKRRR$_{741}$) in the DNA binding (DB) domain (aa 734−761)[18]. This dsDNA binding activity is essential for *Tb*RAP1's association with the telomere chromatin and *Tb*RAP1-mediated *VSG* silencing. However, we do not understand the underlying mechanism of how *Tb*RAP1 selectively permits the active *VSG* to be fully expressed while silencing other *VSGs*.

Here our NMR studies identify an RNA Recognition Motif (RRM) in *Tb*RAP1, while known RAP1 homologs have not been reported to have any RRM domains. Assisted by the R/K patch, *Tb*RAP1 RRM binds the 16-mer consensus sequence of *VSG* 3'UTRs[19,20] in vitro, while *Tb*RAP1 binds the active *VSG* RNA in vivo. Strikingly, mutations in the RRM domain that specifically abolish the *Tb*RAP1-*VSG* RNA interaction lead to an acute decrease in the active *VSG* RNA level by ~50% and subsequent derepression of all silent *VSGs*, thus disrupting both aspects of *VSG* MAE. In contrast, mutations in the R/K patch alone or in both the R/K patch and RRM lead to acute depression of silent *VSGs*, but the active *VSG* RNA is only moderately decreased by ~13%. Mechanistically, *Tb*RAP1's RNA and dsDNA binding activities compete in a substrate concentration-dependent manner. Such competition suggests a mechanism of *VSG* MAE where the active *VSG's* abundant RNA antagonizes *Tb*RAP1's dsDNA

binding-mediated silencing effect at the active *VSG* locus to sustain its full-level expression.

## Results

### *Tb*RAP1 interacts with the active *VSG* RNA in vivo

RAP1 homologs have been identified from kinetoplastids to mammals[21]. None of the known RAP1 homologs has been reported to have any RNA binding activity. We previously found that *Tb*RAP1 does not bind the telomeric repeat-containing RNA (TERRA) in RNA IP experiments[22-24]. However, to our great surprise, we found that *Tb*RAP1 interacts with the active *VSG* RNA (Fig. 1a, b). RNA crosslinking immunoprecipitation (RNA CLIP) assays were performed in *Tb*RAP1$^{F2H}$ $^{+/-}$ cells that express VSG2 as the major surface antigen (Table 1), in which one *Tb*RAP1 allele is deleted and the other has an N-terminal FLAG-HA-HA (F2H) tag[17]. Quantitative RT-PCR (qRT-PCR) analysis detected significantly more active *VSG2* RNA in the *Tb*RAP1 CLIP product than in the negative control IgG CLIP product (Fig. 1a). RNAs of the telomerase reverse transcriptase (*Tb*TERT[25]), small nuclear RNA gene activation protein 50 (SNAP50), and Protein Kinase A catalytic subunit (PKAC1) were also examined in CLIP products. Approximately the same amount of *Tb*TERT, *SNAP50*, and *PKAC1* RNAs were detected in both *Tb*RAP1 and IgG CLIP products (Fig. 1a). Therefore, *Tb*RAP1 interacts with the active *VSG* RNA but not *Tb*TERT, *SNAP50*, or *PKAC1* RNAs. We also performed RNA CLIP in PVS3-2/OD1-1 cells (Table 1) that express VSG9 as the major surface antigen[15]. Again, qRT-PCR detected significantly more *VSG9* RNA in the *Tb*RAP1 CLIP product than in the IgG control (Fig. 1b), indicating that *Tb*RAP1 can interact with the active *VSG* RNA regardless of which *VSG* is expressed. Therefore, we report for the first time that *Tb*RAP1 is associated with the active *VSG* RNA in vivo, an unprecedented finding for RAP1 homologs.

### The *Tb*RAP1 MybLike domain contains an RRM module

To investigate whether the *Tb*RAP1 MybLike domain (aa 639−761) is responsible for binding to the active *VSG* RNA, we first determined the solution structure of *Tb*RAP1$_{639-761}$ by NMR spectroscopy (Fig. 1c, Supplementary Fig. 1a, Supplementary Table 1). The N-terminal region of *Tb*RAP1$_{639-761}$ does not adopt a typical Myb fold but forms a

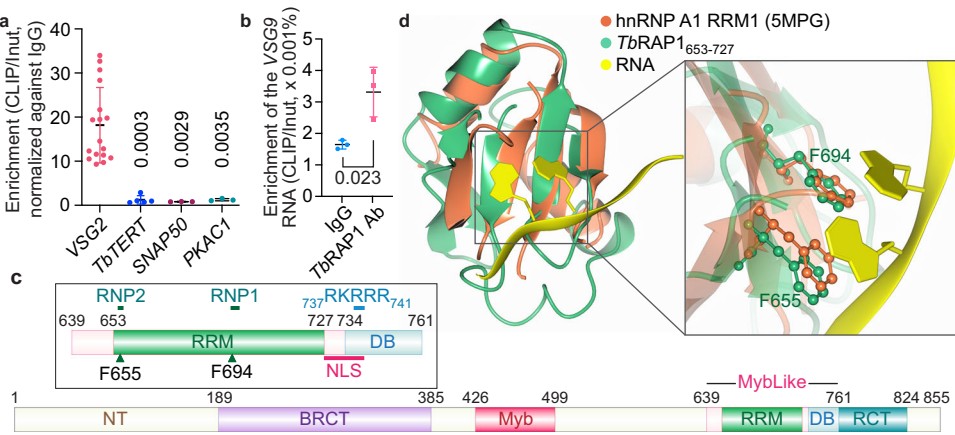

**Fig. 1 | *Tb*RAP1 binds the active *VSG* RNA in vivo and contains an RNA Recognition Motif (RRM) domain. a** RNA CLIP experiments were performed in *Tb*RAP1$^{F2H+/-}$ cells that express VSG2. qRT-PCR was performed to estimate the amount of the *VSG2* RNA and the *Tb*TERT, *SNAP50*, and *PKAC1* RNAs in the RNA CLIP product. Enrichment of the *VSG2*, *Tb*TERT, *SNAP50*, and *PKAC1* RNAs (CLIP/Input) was calculated for the CLIP experiment using the HA antibody 12CA5 and that using IgG. Relative enrichment was calculated using the enrichment of IgG CLIP as a reference. Average and standard deviation were calculated from three (*SNAP50* & *PKAC1*), five (*Tb*TERT), and seventeen (*VSG2*) independent experiments. *P* values of two-sided unpaired *t*-tests (compared to *VSG2* RNA enrichment) are shown. **b** RNA CLIP was performed in VSG9-expressing PVS3-2/OD1-1 cells using a *Tb*RAP1 rabbit

antibody[15] and IgG and the enrichment of the *VSG9* RNA in the CLIP product was calculated. Average and standard deviation were calculated from three independent experiments. Error bars represent standard deviation. Source data are provided as a Source Data file. **c** Domain structure of *Tb*RAP1. Inset, an enlarged diagram of the *Tb*RAP1 MybLike domain (aa 639−761)[15], which contains an RRM (aa 653−727) and the DNA Binding (DB) domain (aa 734−761)[18]. Arrowheads mark the conserved F655 and F694 residues. **d** Superposition of *Tb*RAP1$_{653-727}$ (green) with the RRM1 domain of hnRNP A1 (orange) bound with a short RNA oligo (golden) [https://doi.org/10.2210/pdb5MPG/pdb][28]. Inset highlights that F655 and F694 in *Tb*RAP1 superimpose well with F17 and F59 in hnRNP A1 that form stacking interactions with the RNA substrate.

**Table 1 | List of *T. brucei* strains used in this study[a]**

| Strain | Description | References |
|---|---|---|
| *TbRAP1*[F2H+/-] | One F2H-tagged WT *TbRAP1* and one deleted allele | [17] |
| PVS3-2/OD1-1 | WT *TbRAP1* alleles, VSG9 is active | [15] |
| *TbRAP1*[F/+] | One floxed allele and one WT *TbRAP1* | [17] |
| *TbRAP1*[F/ΔMybL] | One floxed allele and one N-terminally F2H- and NLS-tagged ΔMybL mutant | [17] |
| *TbRAP1*[F/ΔRRM] | One floxed allele and one N-terminally F2H- and NLS-tagged ΔRRM mutant | current study |
| *TbRAP1*[F/2FA&5A] | One floxed allele and one N-terminally F2H- and NLS-tagged F655AF694AR737AK738AR739AR740AR741A mutant | current study |
| *TbRAP1*[F/2FQ] | One floxed allele and one N-terminally F2H-tagged F655QF694Q mutant | current study |
| *TbRAP1*[F/2FL] | One floxed allele and one N-terminally F2H-tagged F655LF694L mutant | current study |
| *TbRAP1*[F/2FA] | One floxed allele and one N-terminally F2H-tagged F655AF694A mutant | current study |
| *TbRAP1*[F/5A] | One floxed allele and one N-terminally F2H- and NLS-.tagged R737AK738AR739AR740AR741A mutant | [18] |
| *TbRAP1*[F/ΔDB] | One floxed allele and one N-terminally F2H- and NLS-tagged ΔDB mutant | [18] |
| *TbRAP1*[/2FL] | One deleted allele and one N-terminally F2H-tagged F655LF694L mutant, derived from *TbRAP1*[F/2FL] cells by treating cells with Cre | current study |

[a]All cells except PVS3-2/OD1-1 express VSG2.

canonical RRM (aa 653–727)[26] with the characteristic topology of a four-stranded anti-parallel β-sheet and two α-helices packed behind the β-sheet (Fig. 1d; Supplementary Fig. 1b). The DB domain (aa 734–761) at the C-terminus of $TbRAP1_{639-761}$ forms a long and flexible loop (Supplementary Fig. 1b, left). In contrast, none of the known RAP1 homologs has been reported to have an RRM domain.

RRM is a conserved structural platform that binds to diverse RNAs and ssDNAs[26,27]. Sequence analysis shows that *Tb*RAP1 RRM contains the signature RNP1 and RNP2 sequence motifs, with F655 in RNP2 and F694 in RNP1 representing the two conserved aromatic residues critical for substrate binding (Supplementary Fig. 1c)[26]. *Tb*RAP1 RRM superimposes well with RRM1 of heterogeneous nuclear ribonucleoprotein (hnRNP) A1 bound with an RNA oligo [https://doi.org/10.2210/pdb5MPG/pdb][28] (Fig. 1d), with a Root Mean Square Deviation (RMSD) of ~3.3-3.5 Å for the main chain atoms. Notably, F655 and F694 of *Tb*RAP1 match exactly to F17 and F59 of hnRNP A1 that form stacking interactions with RNA (Fig. 1d). In addition, sequence alignment and structural prediction by AlphaFold2[29] confirm that RAP1 homologs in representative Trypanosomatida organisms all have a highly conserved RRM (Supplementary Fig. 1d), while vertebrate and fungal RAP1s do not seem to have any RRM domain (Supplementary Fig. 1e). Thus, the RRM domain is uniquely conserved in RAP1 homologs of these microbial parasites but absent in RAP1s from higher eukaryotes.

**TbRAP1 RRM binds to the consensus *VSG* 3'UTR region in vitro**

Since $TbRAP1_{639-761}$ contains an RRM domain plus a flexible DB domain, we then used NMR titration to test whether it binds to the active *VSG* RNA. We used 34-*VSG*-UTR, a 34 nt RNA from the 3'UTR of *VSG2* that contains the consensus 16-mer found in all *VSG* 3'UTRs (Supplementary Table 2)[19,20]. We titrated 34-*VSG*-UTR into $^{15}$N-labeled $TbRAP1_{639-761}$ (Supplementary Table 3) and observed significant concentration-dependent chemical shifts for RNP1 and RNP2 residues, particularly F655 and F694, in heteronuclear single quantum correlation (HSQC) spectra (Fig. 2a–c). A few residues in the DB domain also showed noticeable chemical shifts, although at much lower magnitudes compared to RNP1 and RNP2 residues (Fig. 2a, b). These results suggest that both the RRM module and the DB domain interact with the 34-*VSG*-UTR, with RRM playing a major role. Compared to other RRM domains, the chemical shifts induced by 34-*VSG*-UTR in $TbRAP1_{639-761}$ are mostly in the slow-to-intermediate exchange region, indicative of a moderate micromolar binding affinity[28].

To further characterize how the RRM and DB domains bind RNA, we did similar NMR titration studies using the RRM-containing $TbRAP1_{639-733}$, $TbRAP1_{639-733}$2FL with the two key aromatic residues F655 and F694 of the RRM domain mutated to leucine residues, and $TbRAP1_{639-761}$5A with the R/K patch in the DB domain mutated to five

alanines[18] (Supplementary Table 3). For both $TbRAP1_{639-733}$ and $TbRAP1_{639-761}$, 34-*VSG*-UTR induced similar patterns of chemical shift in RNP1 and RNP2 (Fig. 2, a, b, d, e), but the magnitude was smaller for $TbRAP1_{639-733}$ than for $TbRAP1_{639-761}$ (Fig. 2, b, c, e, f). NMR titration using $TbRAP1_{639-761}$5A also showed similar results as $TbRAP1_{639-733}$ (Supplementary Fig. 1f). However, no chemical shifts were observed for $TbRAP1_{639-733}$2FL even when 34-*VSG*-UTR was in 3-fold molar excess (Fig. 2g). These results indicate that RRM alone can bind 34-*VSG*-UTR, which requires the two conserved aromatic residues F655 and F694, while the DB domain helps to strengthen this binding.

To explore the sequence specificity of *Tb*RAP1 RRM, we tested $TbRAP1_{639-733}$'s binding to (UUAGGG)$_2$, an oligo that contains the TERRA sequence[22–24]. (UUAGGG)$_2$ did not induce any noticeable chemical shifts when titrated to $TbRAP1_{639-733}$ (Fig. 2f; Supplementary Fig. 2a), which is consistent with our previous observation that *Tb*RAP1 does not bind TERRA[24]. We also tested $TbRAP1_{639-733}$'s binding to 35-random, a 35 nt RNA with a random sequence (Supplementary Table 2) by NMR titration. No noticeable chemical shifts were detected, either (Fig. 2f; Supplementary Fig. 2b). These data suggest that $TbRAP1_{639-733}$ does bind RNA with certain sequence specificity.

RRM domains are known to recognize short RNA motifs of 2-8 nucleotides[26,27]. To further map which sequence within 34-*VSG*-UTR can be recognized by *Tb*RAP1 RRM, we performed NMR titration using 16-*VSG*-UTR, an oligo that contains only the 16-mer consensus sequence in *VSG* 3'UTRs (Supplementary Table 2). 16-*VSG*-UTR and 34-*VSG*-UTR induced the same pattern of chemical shifts in both $TbRAP1_{639-761}$ and $TbRAP1_{639-733}$ (Fig. 2a, b, d, e; Supplementary Fig. 2c–f). Therefore, the 16-mer consensus sequence in *VSG* 3'UTRs is sufficient to be recognized by *Tb*RAP1 RRM. In addition, 16-*VSG*-UTR also induced stronger chemical shifts in $TbRAP1_{639-761}$ than $TbRAP1_{639-733}$, further validating the supporting role of the DB domain (Supplementary Fig. 2c–f). Furthermore, the magnitude of chemical shifts induced by 16-*VSG*-UTR for the aromatic residues F655 and F694 in RRM was ~50% lower than those induced by 34-*VSG*-UTR (Supplementary Fig. 2d, f vs. Fig. 2b, e, respectively). These subtle differences suggest that *Tb*RAP1 RRM may recognize additional sequence motifs in the longer 34-*VSG*-UTR substrate, which leads to stronger binding and more prominent chemical shifts. Since RRM domains are known to have promiscuous binding activities, it is likely that *Tb*RAP1 RRM can recognize more than one sequence within the *VSG* RNA.

We also used the fluorescence polarization assay as a biophysical technique to assess the RNA binding activity of *Tb*RAP1. Fluorophore-labeled 16-*VSG*-UTR was titrated to $TbRAP1_{639-761}$, $TbRAP1_{639-733}$, and $TbRAP1_{639-761}$5A and the estimated binding affinity $K_d$ were ~258, 929, and 969 μM, respectively (Supplementary Fig. 2g–i). These data corroborate our NMR studies to

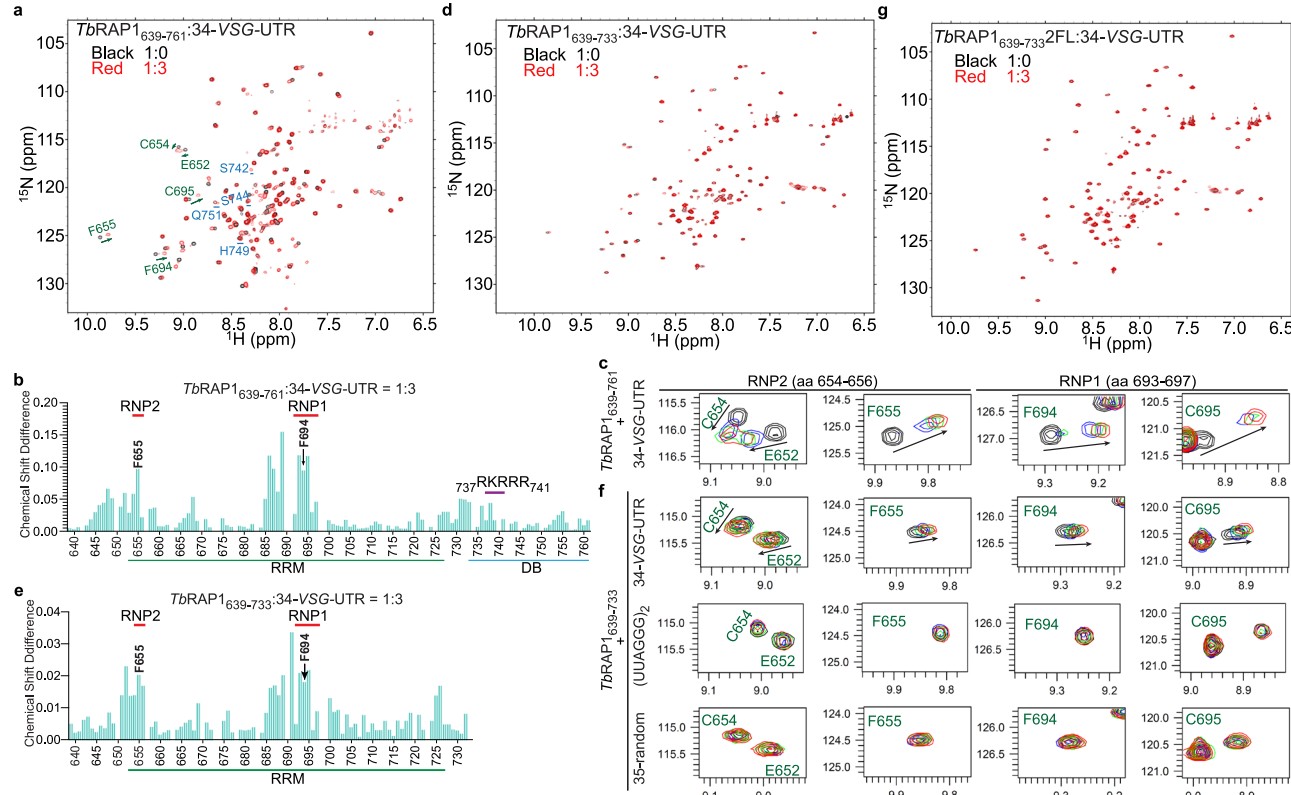

**Fig. 2 | *Tb*RAP1 RRM binds RNAs containing the 16-mer consensus sequence of *VSG* 3'UTRs[20] with a moderate affinity. a, d, g** $^1$H-$^{15}$N HSQC NMR spectra of $^{15}$N-labeled *Tb*RAP1$_{639–761}$ (**a**), *Tb*RAP1$_{639–733}$ (**d**) and *Tb*RAP1$_{639–733}$2FL (**g**) in the absence (black) and presence of 34-*VSG*-UTR in 3× molar excess (red). In (**a**), residues located in the RRM domain (labeled in green) showed noticeable chemical shifts (arrow) while residues in the DB domain (labeled in blue) did not (underline). In (**g**), no chemical shifts were observed. **b, e** Chemical shift differences of individual *Tb*RAP1 residues in NMR titration when *Tb*RAP1$_{639–761}$ (**b**) or *Tb*RAP1$_{639–733}$ (**e**) was used. Source data are provided as a Source Data file. **c** Inset of overlaid $^1$H-

$^{15}$N-HSQC spectra in (**a**) highlighting chemical shift perturbations for key residues in RRM in the absence (black) and presence of 34-*VSG*-UTR in 1x (blue), 2x (green) and 3× (red) molar excess. Residues located on RNP1 and RNP2 of RRM, including the conserved F655 and F694 are highlighted in (**c**). **f** Insets of overlaid $^1$H-$^{15}$N-HSQC spectra of $^{15}$N-labeled *Tb*RAP1$_{639–733}$ in the absence (black) or presence of 34-*VSG*-UTR (top), (UUAGGG)$_2$ (middle), and 35-random (bottom) in 1x (blue), 2x (green) and 3× (red) molar excess. Highlighted residues are the same as in (**c**). Only 34-*VSG*-UTR induced noticeable chemical shifts in the RRM domain. PPM, parts per million.

confirm that *Tb*RAP1 RRM recognizes the 16-mer consensus sequence of *VSG* 3'UTRs. This RNA binding activity requires the two conserved aromatic residues, F655 and F694 in RNP2 and RNP1, respectively, and is enhanced by the DB domain.

We subsequently performed EMSA to validate the *Tb*RAP1 RRM-mediated RNA binding activity. Initially, TrxA-His$_6$ (TH$_6$) or GST-tagged *Tb*RAP1 fragments were used (Supplementary Fig. 3a). TH$_6$-tagged *Tb*RAP1$_{639–761}$, *Tb*RAP1$_{639–733}$, and *Tb*RAP1$_{639–761}$5A (Supplementary Table 3) all bound 170-*VSG*-UTR, a 170 nt RNA containing the *VSG2* 3'UTR sequence (Supplementary Table 2), while TH$_6$ alone or TH$_6$-*Tb*RAP1$_{639–761}$2FA&5A (F655AF694A, $_{737}$RKRRR$_{741}$ mutated to $_{737}$AAAAA$_{741}$, Supplementary Table 3) did not (Supplementary Fig. 3b–d). In addition, GST-*Tb*RAP1$_{414–855}$ bound this RNA, while GST alone and the GST-tagged duplex telomere DNA-binding *Tb*TRF[30] did not (Supplementary Fig. 3b, c, e; Supplementary Table 3).

To examine *Tb*RAP1-specific RNA binding activity without any possible interference by the fusion tag, we cleaved the TH$_6$ tag by 3C and purified tagless *Tb*RAP1 fragments (Supplementary Fig. 3f). Both *Tb*RAP1$_{639–761}$ and *Tb*RAP1$_{639–733}$ bound 170-*VSG*-UTR (Fig. 3a, b) but *Tb*RAP1$_{639–733}$2FQ (F655QF694Q), *Tb*RAP1$_{639–733}$2FL (F655LF694L), or *Tb*RAP1$_{639–733}$2FA (F655AF694A) did not (Fig. 3c, d; Supplementary Fig. 3g; Supplementary Table 3). In addition, more than one *Tb*RAP1$_{639–733}$ molecule can bind the same 170-*VSG*-UTR substrate when the protein:RNA ratio is increased (Fig. 3b).

Unexpectedly, tagless *Tb*RAP1$_{639–761}$ and *Tb*RAP1$_{639–733}$ bound 170-no-*VSG* (Fig. 3e, f; Supplementary Table 2) but none of the

*Tb*RAP1$_{639–733}$2FQ, *Tb*RAP1$_{639–733}$2FL, or *Tb*RAP1$_{639–733}$2FA did (Fig. 3g, h; Supplementary Fig. 3g). Similarly, TH$_6$-tagged *Tb*RAP1$_{639–761}$, *Tb*RAP1$_{639–733}$, and *Tb*RAP1$_{639–761}$5A also bound 170-no-*VSG* (Supplementary Fig. 3h, i). *Tb*RAP1$_{639–733}$ did exhibit higher affinity to 170-*VSG*-UTR than to 170-no-*VSG* (Supplementary Fig. 3j), indicating that *Tb*RAP1 RRM prefers the *VSG* 3'UTR sequence. Nevertheless, the observation that *Tb*RAP1$_{639–733}$ bound 170-no-*VSG* (Fig. 3f) seems inconsistent with the fact that *Tb*RAP1$_{639–733}$ does not bind 35-random in NMR titration (Fig. 2f; Supplementary Fig. 2b). We, therefore, examined whether *Tb*RAP1$_{639–733}$ binds 35-random in EMSA. 35-*VSG*-UTR was used as a positive control, which contains both the 9-mer and the 16-mer consensus motifs in *VSG* 3'UTR (Supplementary Table 2)[20]. *Tb*RAP1$_{639–733}$ bound 35-*VSG*-UTR but not 35-random in EMSA (Fig. 3i), confirming the NMR titration result. RRM domains usually recognize a short RNA sequence of 2-8 nucleotides[26,27]. It is possible that 170-no-*VSG* may contain additional sequences that can be recognized by *Tb*RAP1 RRM other than the consensus sequences in *VSG* 3'UTRs.

We further performed EMSA using the shorter 16-*VSG*-UTR substrate (Supplementary Table 2), to better explore the sequence specificity of *Tb*RAP1's RNA binding activity. Interestingly, *Tb*RAP1$_{639–761}$ clearly bound 16-*VSG*-UTR (Fig. 3j) but *Tb*RAP1$_{639–733}$'s binding affinity appears to be too weak to be detected by EMSA. This observation supports our NMR titration results and further validates the importance of the DB domain in the RRM-mediated RNA binding. Additionally, $K_d$ values estimated by EMSA show stronger affinity of

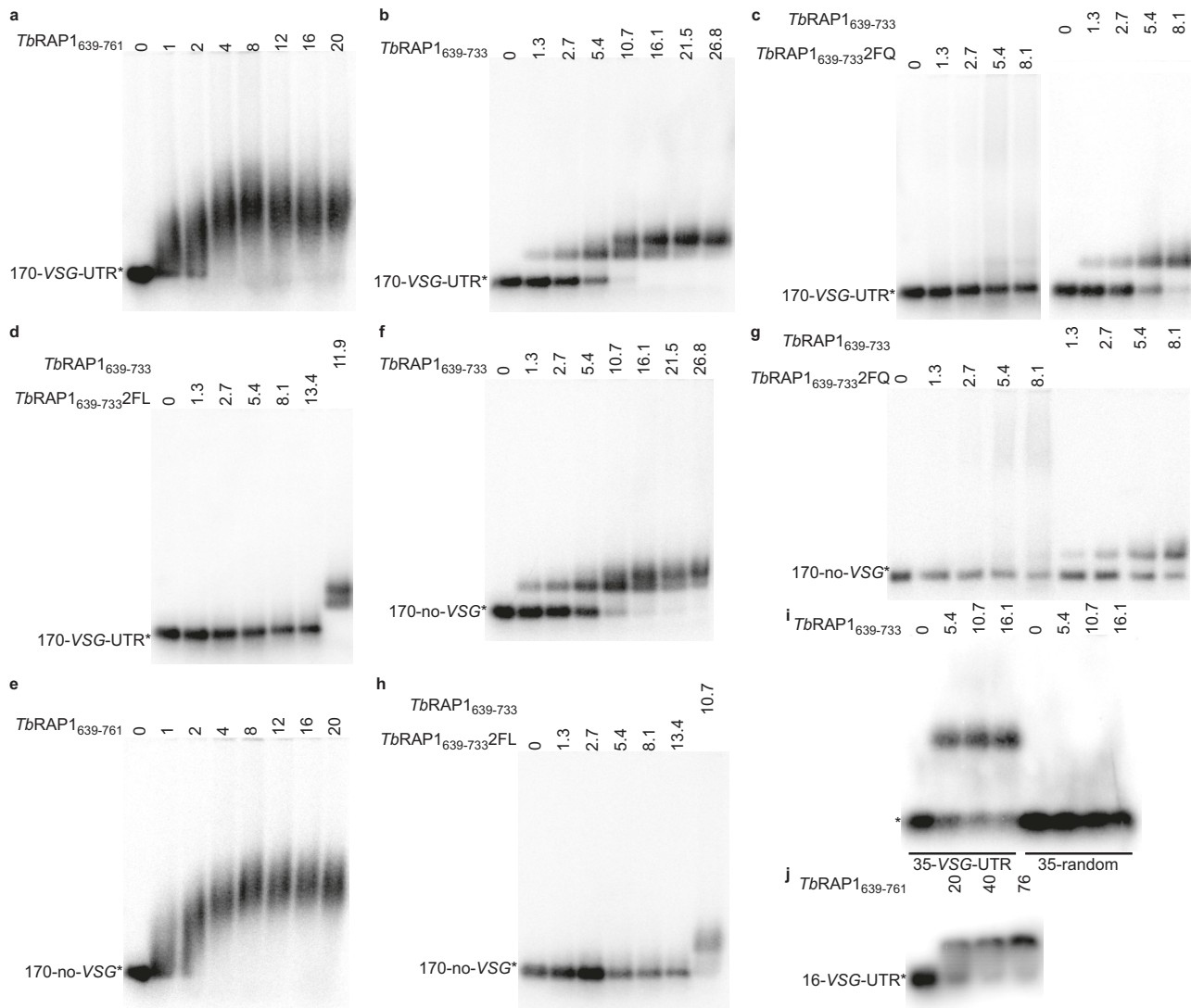

**Fig. 3 | Characterization of *Tb*RAP1 RRM's RNA binding activity by EMSA.** Untagged recombinant *Tb*RAP1$_{639-761}$ (**a**, **e**, **j**), *Tb*RAP1$_{639-733}$ (**b–d**, **f–i**), *Tb*RAP1$_{639-733}$2FQ (**c**, **g**), and *Tb*RAP1$_{639-733}$2FL (**d**, **h**) were incubated with 170-*VSG*-UTR (**a–d**), 170-no-*VSG* (**e–h**), 35-*VSG*-UTR (**i**), 35-random (**i**), or 16-*VSG*-UTR (**j**) (Supplementary Table 2). The concentration of protein (μM) used in each reaction is indicated on top of each lane. Samples were electrophoresed in 0.8% agarose gels (**a–i**) or a 1.2% agarose gel (**j**) in 0.5 x TBE buffer. Source data are provided as a Source Data file.

*Tb*RAP1$_{639-761}$ for 35-*VSG*-UTR than 16-*VSG*-UTR, which is consistent with our NMR titration results (Supplementary Fig. 2j).

## The in vivo *Tb*RAP1-*VSG* RNA interaction depends on the conserved aromatic residues in RRM

We generated *Tb*RAP1$^{F/mut}$ strains by replacing the WT *Tb*RAP1 allele with various RRM mutants in *Tb*RAP1$^{F/+}$ cells (Supplementary Fig. 4a, b; Table 1)[17]. To specifically examine the in vivo RNA binding activities of *Tb*RAP1 mutants, we did RNA CLIP after deleting the loxP-flanked *Tb*RAP1 (the *F* allele) by Cre, as RRM mutants can interact with WT *Tb*RAP1 through its BRCT domain[17]. Removal of the *Tb*RAP1 *F* allele was confirmed by PCR (Supplementary Fig. 4c–g). *Tb*RAP1ΔRRM and *Tb*RAP1ΔMybL (MybLike deletion)[18] were expressed at a subtly lower level than WT *Tb*RAP1 (Supplementary Fig. 4h), while *Tb*RAP1-2FQ, *Tb*RAP1-2FL, *Tb*RAP1-2FA, and *Tb*RAP1-2FA&5A were expressed the same as *Tb*RAP1 (Supplementary Fig. 4i–l). As expected, *Tb*RAP1ΔMybL and *Tb*RAP1ΔRRM mutants that lack the whole RRM domain lost the *Tb*RAP1-*VSG2* RNA interaction (Fig. 4a). Similarly, *Tb*RAP1-2FQ, *Tb*RAP1-2FA, and *Tb*RAP1-2FA&5A did not bind *VSG2* RNA, either (Fig. 4a). Interestingly, although *Tb*RAP1-2FL bound significantly lower amount of *VSG2* RNA than WT *Tb*RAP1, this

mutant appeared to have a smaller RNA binding defect than other mutants (Fig. 4a).

Because *Tb*RAP1 DB enhances the RNA binding activity in vitro, we further examined the effect of DB domain mutations on *VSG* RNA binding in vivo. We previously reported that *Tb*RAP1ΔDB and *Tb*RAP1-5A were expressed at the same level as *Tb*RAP1[18]. Surprisingly, both *Tb*RAP1ΔDB and *Tb*RAP1-5A only pulled down background level of *VSG2* RNA (Fig. 4a). Since neither mutant is associated with the telomere chromatin[18], this observation suggests that being localized at the telomere is a prerequisite for *Tb*RAP1 to bind the active *VSG* RNA, which has a high concentration only at the active *VSG* locus.

We also performed Chromatin IP (ChIP) to test whether the RRM domain is necessary for *Tb*RAP1's localization to the telomere. *Tb*RAP1-2FA&5A did not associate with the telomere chromatin (Fig. 4b), presumably because the 5A mutation already abolished *Tb*RAP1's DNA binding activities[18]. In contrast, *Tb*RAP1-2FQ, *Tb*RAP1-2FL, and *Tb*RAP1-2FA still associated with the telomere chromatin (Fig. 4c, d; Supplementary Fig. 4m). Immunofluorescence (IF) analysis further showed that both *Tb*RAP1-2FQ and 2FL were partially colocalized with *Tb*TRF that binds the duplex telomere DNA[30] the same way as WT *Tb*RAP1 (Fig. 4e). Hence, in vivo binding of *Tb*RAP1 to the active *VSG* RNA

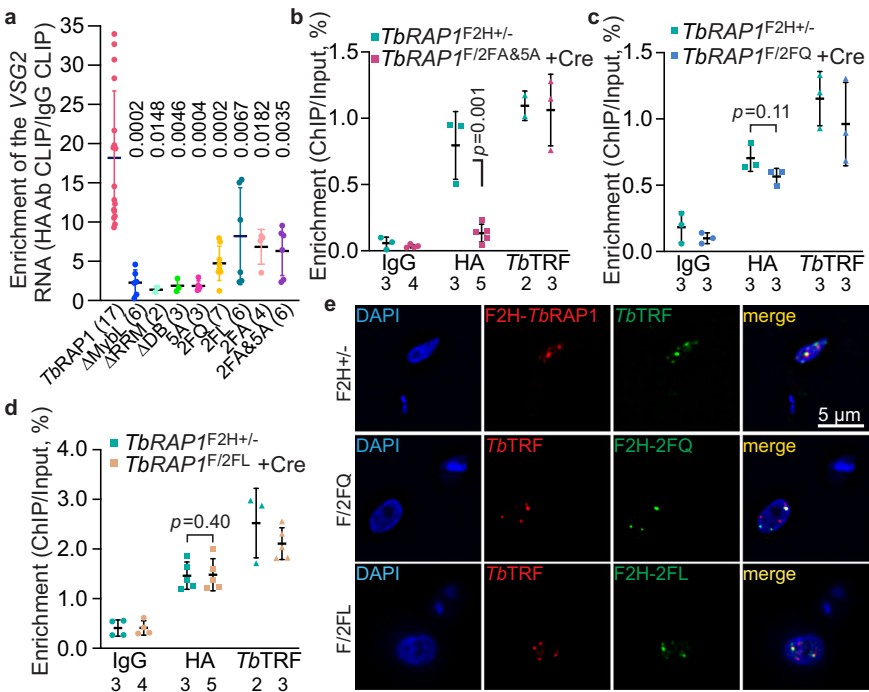

**Fig. 4 | *Tb*RAP1 interacts with the active *VSG* RNA through its RRM domain in vivo. a** RNA CLIP experiments were performed in various *TbRAP1*$^{F/mut}$ strains (expressing VSG2) after a 30-h induction of Cre. The presence of the active *VSG2* RNA in the RNA CLIP product was determined by qRT-PCR. The enrichment of *VSG2* RNA (CLIP/Input) was calculated for the CLIP experiment using the HA antibody 12CA5 and that using IgG. Relative enrichment was calculated using the enrichment of IgG CLIP as a reference. Average and standard deviation were calculated from two to seventeen independent experiments (the exact number of experiments was indicated in parentheses following each strain name). *P* values of two-sided unpaired *t*-tests between the *TbRAP1*$^{F2H+/-}$ and *TbRAP1*$^{F/mut}$ are shown on top of corresponding columns. Data for WT *Tb*RAP1 is the same as that in Fig. 1a. **b**–**d** ChIP experiments using the HA antibody 12CA5, a *Tb*TRF rabbit antibody[30], and IgG were done in *TbRAP1*$^{F2H+/-}$ cells and Cre-induced (for 30 h) *TbRAP1*$^{F/2FA&5A}$ (**b**) *TbRAP1*$^{F/2FQ}$ (**c**) and *TbRAP1*$^{F/2FL}$ (**d**) cells. Average and standard deviation were calculated from two to five independent experiments (exact number of samples are indicated beneath bottom labels). *P* values of two-sided unpaired *t*-tests are shown (ChIP using 12CA5, *TbRAP1*$^{F/mut}$ vs *TbRAP1*$^{F2H+/-}$). Source data are provided as a Source Data file. **e** IF analyses were done in *TbRAP1*$^{F2H+/-}$ (top), *TbRAP1*$^{F/2FQ}$ (middle), and *TbRAP1*$^{F/2FL}$ (bottom) cells. 12CA5 and a *Tb*TRF chicken antibody[15] were used. *TbRAP1* genotypes are listed on the left. DNA was stained by DAPI. All images are of the same scale and a size bar is shown in one of the images.

depends on RRM's two conserved residues F655 and F694 and the R/K patch within the DB domain. Additionally, the RRM-mediated RNA binding activity is not required for *Tb*RAP1's association to the telomere chromatin.

## *Tb*RAP1's RNA binding activity is important for *VSG* MAE and telomere integrity

We examined phenotypes of *TbRAP1*$^{F/ΔRRM}$, *TbRAP1*$^{F/2FQ}$, *TbRAP1*$^{F/2FL}$, *TbRAP1*$^{F/2FA}$, and *TbRAP1*$^{F/2FA&5A}$ after a 30–48 h Cre induction (Supplementary Fig. 5a–e). In *TbRAP1*$^{F/ΔRRM}$, *TbRAP1*$^{F/2FQ}$, *TbRAP1*$^{F/2FA}$, and *TbRAP1*$^{F/2FA&5A}$ cells, Cre induction led to an acute growth arrest (Supplementary Fig. 5f–i). However, *TbRAP1*$^{F/2FL}$ cells showed a slower but not arrested growth phenotype upon Cre induction (Supplementary Fig. 5j), which is consistent with the observation that the 2FL mutant affects the RNA binding less than 2FQ, 2FA, and ΔRRM (Fig. 4a). In addition, substituting an aromatic ring in the phenylalanine residue with a long hydrophobic chain in the leucine residue likely has a weaker effect than substituting it with an alanine.

*VSG* MAE has two essential aspects: silencing all but one *VSGs* and a full-level expression of the active *VSG*. In *TbRAP1*$^{ΔRRM}$, *TbRAP1*$^{2FQ}$, and *TbRAP1*$^{2FL}$ mutants, qRT-PCR analysis after the 30-48 h Cre induction detected a significant decrease (~40-60%) in the active *VSG2* RNA level, while RNA levels of silent *VSGs* increased several hundred-fold (Supplementary Fig. 5k–m), indicating that *Tb*RAP1 RRM is essential for both aspects of *VSG* MAE. The decrease in *VSG2* level is particularly striking because the active *VSG* RNA is ~10,000 fold more abundant than any silent *VSG* RNA (Fig. 5a)[16]. Thus, ~50% reduction of the active *VSG2* RNA represents a more overwhelming change than the several

hundred-fold increase in RNA levels of silent *VSGs*. This decrease is also in distinct contrast to the phenotype of *TbRAP1*$^{F/5A}$ and *TbRAP1*$^{F/ΔDB}$ cells that mutated the R/K patch, where silent *VSGs* were similarly derepressed but the active *VSG* RNA remained at ~90% of the WT level (Supplementary Fig. 5n, o)[18]. Interestingly, in *TbRAP1*$^{2FA&5A}$ mutant, the active *VSG* RNA level was also only decreased to ~87% of the WT level (Supplementary Fig. 5p). Hence, *Tb*RAP1's RNA binding activity is particularly essential for keeping the active *VSG* fully transcribed, while mutating the R/K patch leads to a global *VSG* derepression and renders the *Tb*RAP1-*VSG* RNA interaction unimportant.

We further examined the RNA levels of the active *VSG2* at early time points of 12–36 h after Cre induction in *TbRAP1*$^{F/mut}$ cells, aiming to assess direct effects of *Tb*RAP1 RRM mutations on *VSG* expression. Western analysis confirmed the decrease of the total *Tb*RAP1 level in these cells (Supplementary Fig. 6a–e). Strikingly, the active *VSG2* RNA level showed significant drop by 12 h and continued to decrease over time, dropping to 58%, 68%, and 50% of the WT level by 24 h in *TbRAP1*$^{F/2FQ}$, *TbRAP1*$^{F/2FL}$, and *TbRAP1*$^{F/2FA}$ cells, respectively (Fig. 5b–d, f). In contrast, the *VSG2* RNA level remained close to the WT level in *TbRAP1*$^{F/5A}$ (~90% by 30 h after Cre induction) and *TbRAP1*$^{F/2FA&5A}$ cells (~87% by 36 h after Cre induction) (Fig. 5e–g). Our temporal profiling of the *VSG2* RNA level further confirms that *Tb*RAP1 RRM is critical for sustaining full-level expression of the active *VSG*.

We also examined the RNA levels of several silent *VSGs* at the time points of 12–36 h after Cre induction in *TbRAP1*$^{F/mut}$ cells. Notably, derepression of silent *VSG 3, 6,* and *9* at 12 h after the Cre induction in RRM point mutants was only ~10 fold, significantly milder than the ~100 fold observed in *TbRAP1*$^{F/5A}$ cells (Fig. 5b–e). The magnitude of

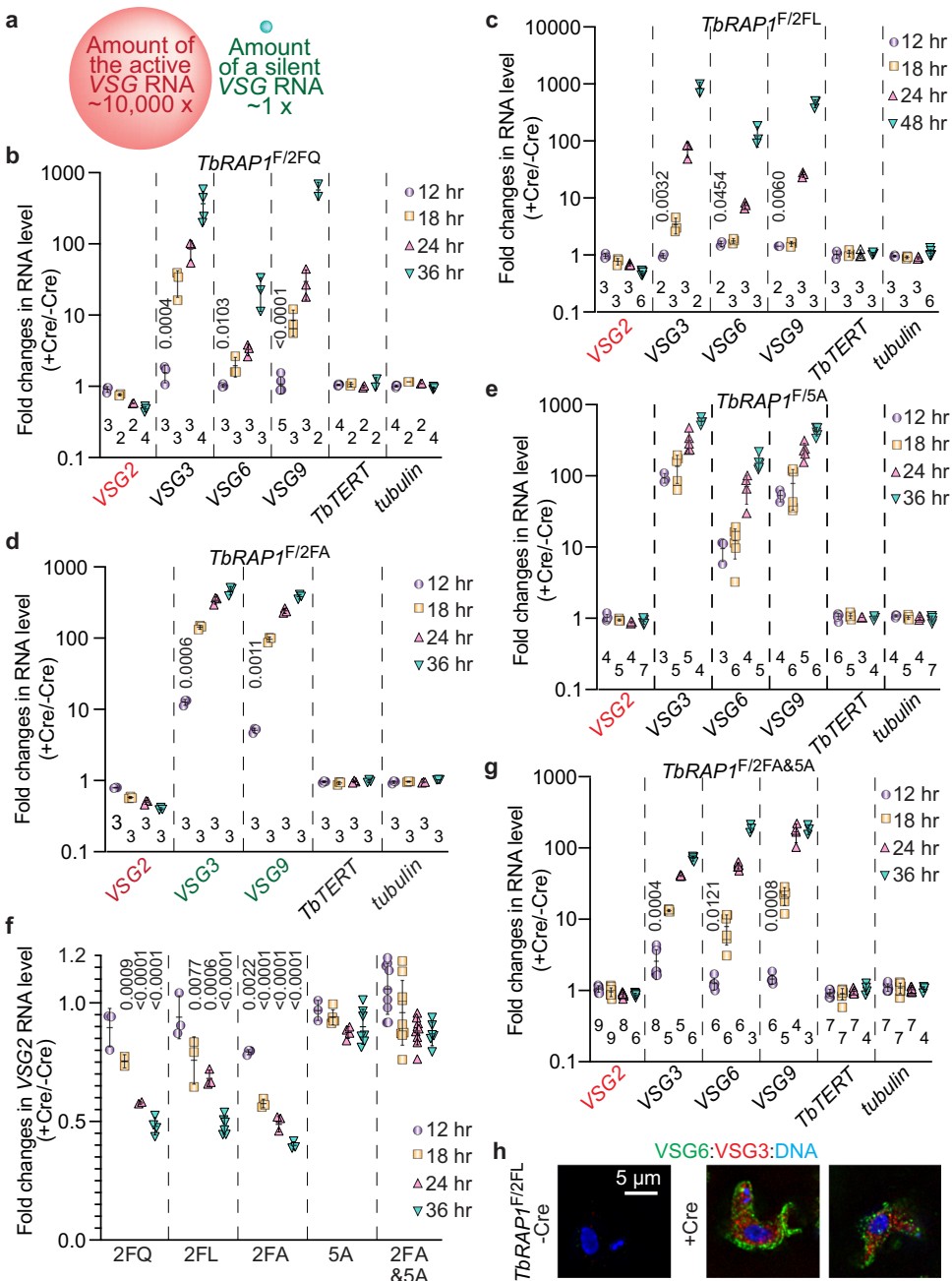

**Fig. 5 | *Tb*RAP1 RRM is essential for full-level expression of the active *VSG*. a** A diagram illustrating the ~10,000-fold difference between the active *VSG* RNA amount and any silent *VSG* RNA amount. Spheres are not drawn to scale. **b**–**e**, **g** qRT-PCR of RNA levels of the active *VSG2* (indicated in red), several silent ES-linked *VSGs*, and chromosome internal *TbTERT* and *tubulin* in *TbRAP1*^F/2FQ (**b**), *TbRAP1*^F/2FL (**c**), *TbRAP1*^F/2FA (**d**), *TbRAP1*^F/5A (**e**), and *TbRAP1*^F/2FA&5A (**g**) cells. The fold changes in RNA level are shown in the log scale. Average and standard deviation were calculated from two to nine independent experiments (exact number of samples are indicated beneath each column). The change in *VSG2* RNA level in these mutants is plotted again in the linear scale in (**f**). At the 12 h point, derepression of *VSG3*, 6, and

9 in *TbRAP1*^F/2FQ, *TbRAP1*^F/2FL, *TbRAP1*^F/2FA, and *TbRAP1*^F/2FA&5A cells was compared to that in *TbRAP1*^F/5A by two-sided unpaired student *t*-tests, and *p* values of significant differences are indicated on top of corresponding columns in (**b**–**d**, **g**). The changes in the *VSG2* RNA level at all time points were compared to that in *TbRAP1*^F/5A cells in the same way. *P* values of significant differences are indicated on top of corresponding columns in (**f**). Error bars in (**b**–**g**) represent standard deviation. Source data are provided as a Source Data file. **h** IF analysis of *TbRAP1*^F/2FL cells before and after the Cre induction. Antibodies specifically recognizing VSG6 (green) and VSG3 (red), which were silent in WT cells, were used. DAPI was used to stain DNA. All panels are of the same scale, and a size bar is shown in one of the panels.

depression became similar at later time points of 18, 24, and 36 h (Fig. 5b–e). Nevertheless, both VSG3 and VSG6, two silent VSGs in uninduced *TbRAP1*^F/2FL cells, were expressed simultaneously in individual cells upon Cre induction (Fig. 5h). Overall, these results confirm that disrupting *Tb*RAP1's RNA binding indeed led to *VSG* derepression, albeit with a slower kinetic profile compared to mutations in the DB domain.

Subsequently, we examined the transcriptome profiles in *TbRAP1*^F/2FQ and *TbRAP1*^F/2FA&5A cells by RNAseq. ~5,000 genes were upregulated and 200-1500 genes were down-regulated in the *TbRAP1*^F/2FQ and *TbRAP1*^F/2FA&5A cells (Supplementary Fig. 6f, g). A large number of *VSG* genes and pseudogenes were up-regulated in both mutants, including all silent *VSGs* in bloodstream form *VSG* ESs (Supplementary Figs. 7a, b and 8). GO term analysis indicated that significantly

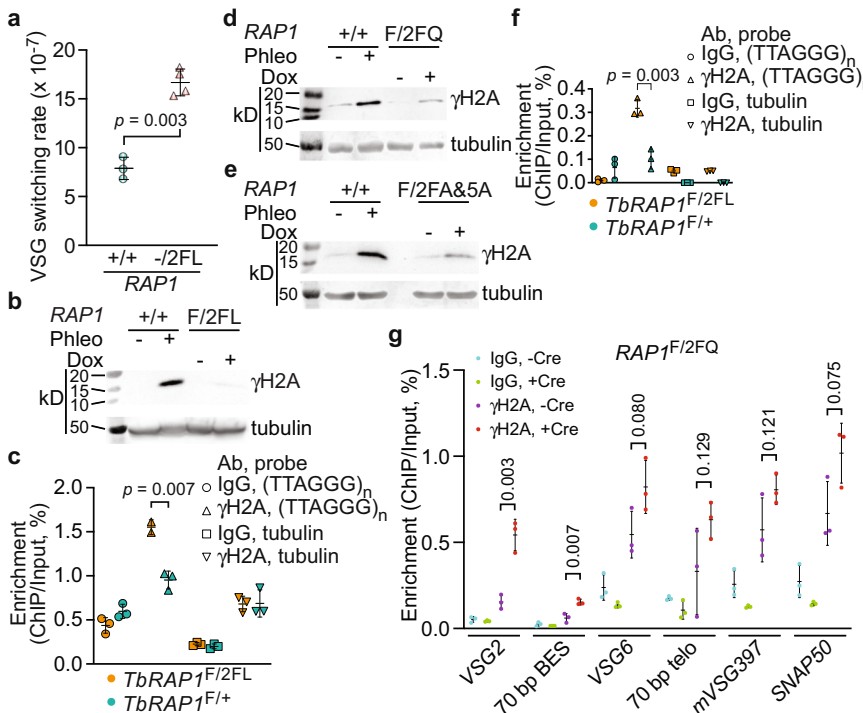

**Fig. 6 | *Tb*RAP1 RRM mutants have an increased amount of DNA damage at the telomere and the subtelomere. a** *TbRAP1$^{-/2FL}$* exhibits an increased VSG switching rate. Average and standard deviation were calculated from three (WT) and four (*TbRAP1$^{-/2FL}$*) independent experiments. *P* values of two-sided unpaired *t*-tests are shown (*TbRAP1$^{-/2FL}$* vs *TbRAP1$^{+/+}$*). **b, d, e** Western analyses to examine the γH2A protein level in WT cells before and after phleomycin treatment (as a positive control) and in *TbRAP1$^{F/2FL}$* (**b**), *TbRAP1$^{F/2FQ}$* (**d**), and *TbRAP1$^{F/2FA&5A}$* (**e**) cells before and after a 30–48 h Cre induction. A γH2A rabbit antibody[23] and the tubulin antibody TAT-1[49] were used. Molecular marker was run on the left lane in each gel and their sizes are indicated on the left. **c, f** ChIP using the γH2A rabbit antibody and IgG in *TbRAP1$^{F/2FL}$* (**c**) and *TbRAP1$^{F/2FQ}$* (**f**) cells after a 30 h Cre induction followed by Southern blotting using a telomere and a tubulin probe. Blots were exposed to a phosphorimager. Images were quantified using ImageQuant and average and standard deviation were calculated from two (γH2A antibody, (TTAGGG)$_n$ probe in *TbRAP1$^{F/2FL}$* cells) or three (all other samples) independent samples in (**c**) and three independent experiments in (**f**). *P* values of two-sided unpaired *t*-tests (mutant vs. control cells) are shown. **g** ChIP using a γH2A rabbit antibody and IgG in *TbRAP1$^{F/2FQ}$* cells followed by quantitative PCR using primers specific to the indicated active and silent ES loci. *SNAP50* is a chromosome internal gene. Average enrichment (ChIP/Input) was calculated from three independent experiments. *P* values of two-sided unpaired *t*-tests (γH2A ChIP products, +Cre vs. -Cre) are shown. Error bars in (**a, c, f, g**) represent standard deviation. Source data are provided as a Source Data file.

derepressed genes are predominantly involved in host immune response evasion (Supplementary Fig. 7c). We also estimated the *VSG2* RNA half-life in *TbRAP1$^{F/ΔMybL}$*, *TbRAP1$^{F/ΔRRM}$*, and *TbRAP1$^{F/2FQ}$* cells. The *VSG2* RNA levels were examined by qRT-PCR after various lengths of time of Actinomycin D treatment, but the half-life of *VSG2* RNA did not change in RRM mutants (Supplementary Fig. 9).

We previously showed that *Tb*RAP1 suppresses VSG switching by maintaining genome integrity at the telomere and subtelomere[23]. Since *Tb*RAP1-2FL is viable, we estimated the VSG switching rate in *TbRAP1$^{-/2FL}$* cells (Table 1), which is twice as high as that in WT cells (Fig. 6a), suggesting that the *Tb*RAP1's RNA binding activity also helps suppress VSG switching. In addition, the level of γH2A, an indicator of DNA damage[31], was increased mildly (Fig. 6b), and significantly more γH2A was associated with the telomere chromatin in Cre-induced *TbRAP1$^{F/2FL}$* cells (Fig. 6c). *Tb*RAP1ΔRRM, *Tb*RAP1-2FQ, *Tb*RAP1-2FA, and *Tb*RAP1-2FA&5A mutants exhibited a strong growth arrest phenotype (Supplementary Fig. 5f–i), which prevented us from determining the VSG switching rate in these mutants. Therefore, we examined the γH2A levels. An increased level of γH2A was observed in Cre-induced *TbRAP1$^{F/2FQ}$*, *TbRAP1$^{F/2FA&5A}$*, *TbRAP1$^{F/2FA}$*, and *TbRAP1$^{F/ΔRRM}$* cells (Fig. 6d, e; Supplementary Fig. 6h–i), indicating that these mutants had more DNA damage. Particularly, we observed an increased amount of γH2A associated with the telomere chromatin (Fig. 6f) and the active ES (Fig. 6g) in *TbRAP1$^{F/2FQ}$* cells after the Cre induction, indicating that *Tb*RAP1's RNA binding activity is also critical for telomeric and subtelomeric integrity. Telomeric DNA breaks, particularly those at the

active *VSG* vicinity, can lead to cell death in >80% of parasites[32], which can explain why RRM mutants have growth defects.

**TbRAP1 binds DNA and RNA in a mutually exclusive manner**
Our NMR structure of *Tb*RAP1$_{639-761}$ shows that the DB domain forms a long and flexible loop that does not contact the RRM module (Supplementary Fig. 1b). Thus, it is theoretically possible for *Tb*RAP1 to bind DNA and RNA simultaneously. To test this possibility, we conducted EMSA assays using both dsDNA and RNA substrates. We first confirmed that *Tb*RAP1$_{639-761}$ bound a duplex telomeric DNA probe, 100-ds(TTAGGG) (Fig. 7a; Supplementary Table 2)[18]. However, when non-radiolabeled 170-*VSG*-UTR and radiolabeled 100-ds(TTAGGG) were both incubated with *Tb*RAP1$_{639-761}$, no ternary complex of *Tb*RAP1-RNA-DNA was observed (Fig. 7b). Instead, the amount of *Tb*RAP1-DNA complex gradually decreased in the presence of an increasing amount of 170-*VSG*-UTR (Fig. 7b). Similarly, when non-radiolabeled 100-ds(TTAGGG) and radiolabeled 170-*VSG*-UTR were both incubated with *Tb*RAP1$_{639-761}$, no ternary complex was observed while the amount of *Tb*RAP1-RNA gradually decreased with increasing amount of 100-ds(TTAGGG) (Fig. 7c). EMSA estimated that the $K_d$ values for binding either 100-ds(TTAGGG) or 170-*VSG*-UTR by *Tb*RAP1$_{639-761}$ are comparable in the range of ~100–300 nM (Fig. 7d), thus allowing two-way competition. To investigate whether such competition applies to shorter DNA or RNA substrates, we further compared *Tb*RAP1$_{639-761}$ binding on 80-dsDNA and 81-*VSG*-UTR (Supplementary Table 2), as the shortest ssDNA and dsDNA that *Tb*RAP1 can bind is ~60 nt and 60 bp,

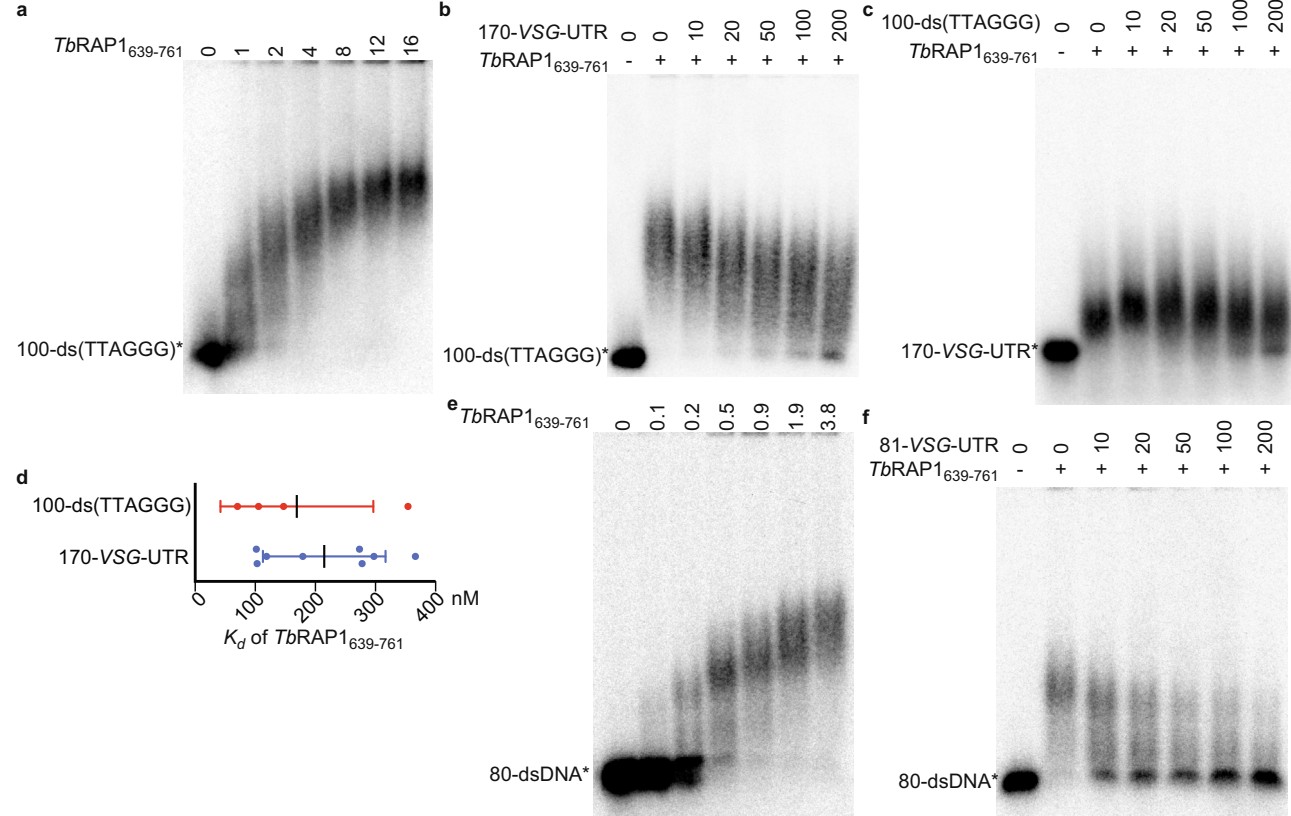

**Fig. 7 | *Tb*RAP1's RNA and DNA binding activities compete with each other.** EMSA experiments were performed using *Tb*RAP1[639–761]. Radiolabeled 100-ds(TTAGGG) (**a**, **b**), 170-*VSG*-UTR (**c**), and 80-dsDNA (**e**, **f**) were used as the binding substrates. Non-radiolabeled 170-*VSG*-UTR (**b**), 100-ds(TTAGGG) (**c**), and 81-*VSG*-UTR (**f**) were used as competitors. The concentration of proteins (μM) used in each experiment is indicated on top of each lane in (**a**) and (**e**). 4.7 μM (**b**), 2.35 μM (**c**), and 0.5 μM (**f**) of *Tb*RAP1[639–761] was used in each competition reaction. The molar excess of the competitor is indicated on top of each lane in (**b**, **c**, **f**). Samples were electrophoresed in 0.8% agarose gels in 0.5x TBE buffer. **d** *Tb*RAP1[639–761]'s affinities to 100-ds(TTAGGG) and 170-*VSG*-UTR ($K_d$ values) were estimated by EMSA. Average and standard deviation were calculated from four (for 100-ds(TTAGGG)) or eight (for 170-*VSG*-UTR) independent experiments. Source data are provided as a Source Data file.

respectively[18]. *Tb*RAP1[639-761] bound 80-dsDNA (Fig. 7e) as expected[18]. When an increasing amount of 81-*VSG*-UTR was added to the reaction using radiolabeled 80-dsDNA as the substrate, no ternary complex of *Tb*RAP1-RNA-DNA was observed, but 81-*VSG*-UTR competed away *Tb*RAP1[639-761]'s binding on 80-dsDNA (Fig. 7f). Therefore, DNA and RNA bind to *Tb*RAP1 in mutually exclusive and competitive manner due to their overlapping binding site and comparable binding affinities.

## Discussion

Our NMR studies reveal that *Tb*RAP1 MybLike folds into an RRM module with a canonical βαββαβ topology, in addition to a C-terminal flexible loop corresponding to the DB domain[18]. Similar to other RRM domains, the four-stranded β-sheet in *Tb*RAP1 RRM is a functional RNA binding site, with conserved residues F655 and F694 poised to form stacking interactions with RNA substrates[26,27]. Hence, *Tb*RAP1 is different from known RAP1 homologs, which do not contain RRM or possess any RNA binding activity. Thus, our study uncovers an additional important function of this essential telomere protein.

We have validated the RNA binding activity of *Tb*RAP1 RRM by NMR titration, fluorescent polarization, and EMSA in vitro, which is also confirmed by RNA CLIP in vivo. In vitro, *Tb*RAP1 RRM alone can bind the 16-mer consensus sequence of *VSG* 3'UTRs[20]. This binding explains why *Tb*RAP1 interacts with all types of active *VSG* RNA in vivo. Moreover, RRM domains are known to be promiscuous and capable of recognizing many RNA sequence motifs. As a result, it is probable that *Tb*RAP1 may bind to RNA sequences aside from the consensus 16-mer of *VSG* 3'UTRs.

Indeed, our NMR titration data shows that *Tb*RAP1 RRM bound to 35-*VSG*-UTR with stronger affinity than 16-*VSG*-UTR, and *Tb*RAP1 RRM also binds a 170-nt long RNA without any *VSG2* 3'UTR sequence in the EMSA experiment. In addition, it is possible that *Tb*RAP1 RRM may recognize the structural features of *VSG* 3'UTRs, as they have been predicted to form a common secondary structure[20]. Future experiments are needed to determine additional RNA sequences, both within and outside *VSG* RNAs, that are recognized by *Tb*RAP1 RRM.

Interestingly, we found that *Tb*RAP1 DB enhances the RRM-mediated RNA binding activity. This effect is particularly significant for recognizing short RNA oligos in vitro, such as 16-*VSG*-UTR. As the DB domain is a long and flexible loop with little inter-domain interaction with RRM, it is possible that RRM and DB contact different parts of the RNA substrate independently. This bi-valent binding mode may achieve higher binding affinity than RRM alone. This combinatorial effect to enhance RNA binding affinity has been reported in other RNA binding proteins. In particular, many RNA splicing proteins such as FUS and hnRNPU contain intrinsically disordered arginine-rich RS or RGG repeats adjacent to well-folded RNA-binding domains such as RRM or Zinc Finger (ZF) domains[33,34]. These disordered motifs are capable of sequence-independent RNA interaction and have been reported to synergize with RRM or ZF to enhance overall RNA binding[34,35]. Additionally, the dual roles of the DB domain in mediating both *Tb*RAP1's DNA and RNA binding offers a mechanistic underpinning for mutually exclusive and concentration-dependent competition between the two activities.

Examination of *VSG* RNA levels reveals an essential role of *Tb*RAP1 RRM in maintaining the full-level expression of the active *VSG*, which is

a critical aspect of *VSG* MAE. We observed a striking decrease of ~50% in the active *VSG* RNA level in RRM point mutants. For two reasons, this decrease most likely did not result from a reduced amount of available RNA polymerase I for transcribing the active *VSG*, even though silent *VSGs* were derepressed globally. Firstly, *Tb*RAP1 DB mutants such as ΔDB and 5A only affect the active *VSG* RNA level subtly, even though silent *VSGs* were derepressed up to ~1000 fold. Secondly, at 12 h post Cre-induction, the active *VSG* RNA level in RRM mutants is already significantly lower than that in the 5A mutant, while silent *VSGs* have not been derepressed to the same extent as that in the 5A mutant. Furthermore, the decreased active *VSG* RNA level in RRM point mutants is not caused by RNA processing, as the half-life of the active *VSG* RNA is not affected by RRM mutations, and *Tb*RAP1 is primarily a nuclear protein as shown in IF analyses (Fig. 4e)[15].

The notable phenotype we observed for active *VSG* is closely related to the moderate binding affinity of the *Tb*RAP1-*VSG* RNA interaction and its competition with the DB-mediated DNA binding activity. Our EMSA studies reveal that *Tb*RAP1's RNA and DNA binding activities are mutually exclusive and compete in a substrate concentration-dependent manner. Thus, the formation of *Tb*RAP1-RNA and *Tb*RAP1-DNA complexes depends on the relative abundance of RNA as opposed to DNA. At silent ESs, *VSGs* are not transcribed, and the *VSG* RNA level is very low. *Tb*RAP1 binds dsDNA by its DB domain and establishes/maintains proper *VSG* silencing (Fig. 8). In contrast, at the active ES, *VSG* is highly transcribed by RNA polymerase I, representing ~10% of total RNA[6,16,36]. As nascent *VSG* RNA is colocalized with the active ES when examined by IF/FISH[37], the local concentration of the active *VSG* RNA is expected to greatly exceed that of the local dsDNA. Hence, at the active *VSG* locus, *Tb*RAP1 binds the *VSG* RNA via its RRM domain instead, disrupting the silencing effect mediated by its dsDNA binding activity (Fig. 8). Therefore, while *Tb*RAP1's dsDNA binding activity silences *VSGs* globally, its RNA binding activity selectively sustains the full-level VSG expression at the active *VSG* locus.

The moderate RNA binding activity is also consistent with the observation that both *Tb*RAP1ΔDB and *Tb*RAP1-5A mutants lose their interaction with the active *VSG* RNA in vivo, although these mutants bind various RNA substrates in vitro. Presumably, a high concentration of the active *VSG* RNA is essential for the *Tb*RAP1-*VSG* RNA interaction. Disrupting *Tb*RAP1's DNA binding activity removes *Tb*RAP1 from the telomere and away from nearby ESs[18], including the active *VSG* locus, the only nuclear location where a high concentration of *VSG* RNA is expected. Thus, *Tb*RAP1ΔDB and *Tb*RAP1-5A mutants likely do not get access to a high concentration of *VSG* RNA and bind it. This prerequisite presumably increases the specificity of *Tb*RAP1's RNA binding activity in vivo, and *Tb*RAP1 is unlikely to interact with random RNA if it is not associated with the chromatin. In addition, in *Tb*RAP1ΔDB and *Tb*RAP1-5A mutants[18], the *Tb*RAP1-*VSG* RNA interaction is no longer

required for a high level of expression of the active *VSG*, as the *Tb*RAP1-mediated silencing effect is abolished in the first place[18]. This further supports our hypothesis that *Tb*RAP1's RNA binding mainly antagonizes *Tb*RAP1's dsDNA binding-mediated silencing effect (Fig. 8).

*Tb*RAP1 RRM mutants are defective in *VSG* silencing even though its dsDNA binding activity is intact. Such *VSG* derepression displayed subtly slower kinetics than in the 5A mutant with disrupted DNA binding. *T. brucei* has been reported to sense the decreased VSG translation and induce *VSG* mRNA synthesis[36]. It is possible that, in *Tb*RAP1 RRM mutants, the decreased level of *VSG* expression induced silent *VSG* derepression through this sensing mechanism. Conversely, the *Tb*RAP1-*VSG* RNA interaction may send a direct signal to allow the silencing of other *VSGs*.

Among examples of MAE, underlying molecular mechanisms are highly process-dependent. In *Borrelia* bacteria[38] and *Pneumocystis* yeast[39], the MAE of their major surface antigen genes requires a unique expression site and is achieved by transcribing only the expression site-resident allele. In *P. falciparum*, histone modification and pairing of *var* gene intron and promotor play important roles in *var* gene MAE[40]. Our study uncovers the competition between *Tb*RAP1's RNA-binding and DNA-binding activities as a mechanism of *VSG* MAE, prompting more detailed investigations and facilitating a deeper understanding of antigenic variation in *T. brucei*. Notably, the RAP1 RRM domain is highly conserved among Trypanosomatids but absent in higher eukaryotes. With this feature and activity likely conserved in Trypanosomatida organisms, *Tb*RAP1 can serve as a promising target for antiparasitic agents.

## Methods

### *T. brucei* strains and plasmids

All *T. brucei* strains used in this study (Table 1) are derived from bloodstream form Lister 427 cells that express the T7 polymerase and the Tet repressor (Single Marker, aka SM)[41]. All strains express VSG2 except that PVS3-2/OD1-1 expresses VSG9[15]. All *T. brucei* cells were cultured in the HMI-9 medium supplemented with 10% FBS and appropriate antibiotics.

*TbRAP1*[F/+] was established previously and described in ref. [17]. All *TbRAP1*[F/mut] strains were established using the same strategy. N-terminal F2H- and NLS-tagged *Tb*RAP1-2FA&5A, *Tb*RAP1-5A, *Tb*RAP1ΔDB, and *Tb*RAP1ΔRRM, F2H-tagged *Tb*RAP1-2FQ, *Tb*RAP1-2FL, and *Tb*RAP1-2FA flanked by sequences upstream and downstream of the *TbRAP1* gene, together with a *PUR* marker, were cloned into pBluescript-SK to generate respective targeting constructs. Mutant targeting plasmids were digested with SacII before transfecting the *TbRAP1*[F/+] cells to generate respective *TbRAP1*[F/mut] strains, which were confirmed by Southern and sequencing analyses.

Bacterial expression plasmids used in this study are listed in Supplementary Table 3.

### Quantitative real-time PCR (qRT-PCR)

For qRT-PCR, total RNA was isolated from *T. brucei* cells using RNAstat-60 (TelTest, Inc.), treated by DNase (Qiagen), and purified using the RNeasy column (Qiagen). cDNA was synthesized using a random hexamer and the MMLV reverse transcriptase (Promega) according to the manufacturer's manual. cDNA and γH2A ChIP product were analyzed by real-time PCR on a CFX Connect (Bio-Rad) using SsoAdvanced Universal SYBR® Green Supermix (Bio-Rad) according to the manufacturer's manual. rRNA level was measured and used as a loading control. Data acquired on CFX Connect were processed using MS Excel and Graphpad Prism. qPCR primer sequences are listed in Supplementary Table 4.

### Chromatin Immunoprecipitation (ChIP)

200 million cells were cross-linked by 1% formaldehyde for 20 min at RT with constant mixing, and the cross-linking was stopped by 0.1 M Glycine. Chromatin was sonicated by a BioRuptor for 6 cycles (each 30 sec

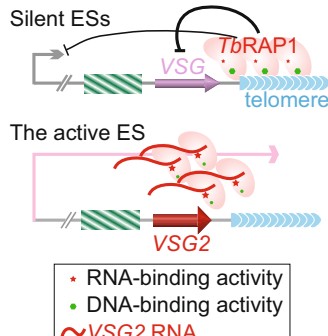

**Fig. 8 | A tentative model for the function of *Tb*RAP1 RRM-*VSG* RNA interaction.** Dark lines with terminal bars represent repressive effect. Thicker line represents stronger effect. Red curved lines represent nascent *VSG2* RNA. Red star and green hexagon represent the RNA and dsDNA binding activities of *Tb*RAP1, respectively. Binding *VSG2* RNA competes *Tb*RAP1's ability to bind local telomeric DNA.

sonication and 30 sec rest) at the high level to get DNA fragments of ~500 bp on average. After saving a small amount of sonicated sample as the input fraction, the sample was equally divided into three fractions, each incubating with 1 μg of HA monoclonal antibody 12CA5 (MSKCC Antibody & Bioresource Core), *Tb*TRF rabbit antibody[30], or IgG conjugated with Dynabeads-Protein G (ThermoFisher) for three hs at 4 °C. In γH2A ChIP, the total lysate was equally divided into two fractions, each incubating with 1 μg of γH2A rabbit antibody[23] or IgG conjugated with Dynabeads-Protein G. After washing, IPed products were eluted from the beads and DNA was isolated from the products followed by Southern slot blot hybridization or quantitative PCR analysis.

### Recombinant protein expression and purification
All recombinant proteins used in this study are listed in Supplementary Table 3.

Recombinant protein expression constructs were transformed into various *E. coli* strains for optimum expression (Supplementary Table 3). Protein samples used for EMSA studies were expressed in standard LB media. Proteins used for acquiring $^{15}$N HSQC NMR spectrum were expressed in M9 minimal media, with $^{15}$N labeled ammonium chloride ($^{15}$N,98%+) (Cambridge Isotope Laboratories, Inc.) as nitrogen source and D-Glucose (Cambridge Isotope Laboratories, Inc.) as carbon source. Protein expression was induced by IPTG. TrxA-His$_6$ (TH$_6$)-tagged proteins were purified with His•bind resin (Millipore) or NiNTA agarose (Qiagen) according to the manufacturer's protocol. GST-tagged proteins were purified with Glutathione Sepharose 4 Fastflow beads (GE) according to manufacturer's protocol. Purified proteins were dialyzed in dialysis buffer (20 mM HEPES pH 7.9, 100 mM KCl, 0.1 mM EDTA, 1 mM PMSF, 15% Glycerol, and 1 mM DTT) at 4 °C overnight. Affinity purified TH$_6$-tagged proteins were dialyzed in 3C protease reaction buffer (50 mM Tris pH 8, 150 mM NaCl). A total of 4 mg of dialyzed protein was digested with 140 units of Pierce™ HRV 3C Protease (ThermoFisher Scientific) at 4 °C overnight with nutation. The digestion mix was passed through NiNTA agarose column (Qiagen) to remove TH$_6$ and 3C-His$_6$. Tagless protein was collected from the flow-through fraction and concentrated using a Centricon (Millipore).

### Fluorescence polarization
Fluorescence polarization (FP) experiments were conducted in black 384-well microplates with triplicates. To determine the equilibrium dissociation constant ($K_d$) between 16-*VSG*-UTR and each of the recombinant *Tb*RAP1$_{639-761}$, *Tb*RAP1$_{639-761}$5A, and *Tb*RAP1$_{639-733}$ proteins, serial ten two-fold dilutions of the proteins were prepared in the binding buffer (20 mM sodium phosphate, 50 mM KCl, 5% glycerol, pH 7.0). The highest protein concentration started at 1200 μM, and 5′ 6-carboxyfluorescein (6-FAM) labeled 16-*VSG*-UTR (Sigma-Aldrich) was diluted to 50 nM. 40 μl protein-RNA mixtures were incubated for 20 min at room temperature. FP signals were detected at 25 °C using a CLARIOstar (BMG LABTECH, Germany) multi-mode microplate reader equipped with polarization filters with excitation wavelengths at 482 nm (482−16 mode) and emission wavelengths at 530 nm (530−40 mode), respectively. The anisotropy value was obtained with a unit of millipolarization (mP). Focus and gain were adjusted by a reference well containing FAM-labeled RNA only. The $K_d$ value was analyzed by fitting a nonlinear regression curve with one site-specific binding mode in GraphPad Prism.

### Electrophoretic mobility shift assay (EMSA)
Purified recombinant proteins were incubated with 0.3 nM radiolabeled 170-*VSG*-UTR/170-no-*VSG* (Fig. 3a–h; Fig. 7c; Supplementary Fig. 3g), 1.5 nM radiolabeled 35-*VSG*-UTR/35-random (Fig. 3i), 10 nM radiolabeled 16-*VSG*-UTR (Fig. 3j), or 1.2 nM 170-*VSG*-UTR/170-no-*VSG* (Supplementary Fig. 3b−e, h, i) in 15 μl of 1 X RNA EMSA buffer (20 mM HEPES pH 7.9, 235 mM KCl, 1 mM MgCl$_2$, 0.1 mM EDTA, 100 ng/μl BSA, 5% Glycerol, 1 mM DTT) at room temperature for 30 mins. Samples were electrophoresed in a 0.8% agarose (Fig. 3, a–i; Fig. 7; Supplementary Fig. 3e, g), a 1.2% agarose (Fig. 3j), or a 5% native polyacrylamide gel (Supplementary Fig. 3b−d, h, i) in 0.5 X TBE running buffer. Gels were dried and exposed to a phosphorimager.

### Probe preparation for EMSA
RNA probes were in vitro transcribed from 120 ng of template DNA using the Maxiscript T7/SP6 transcription kit (ThermoFisher) according to the manufacturer's protocol. Radiolabeled RNA was gel-purified by 10% denaturing PAGE. Purified RNA was resuspended in 40 μl of RNase-free ddH2O. 35-*VSG*-UTR, 35-random, and 16-*VSG*-UTR (Supplementary Table 2) were synthesized by IDT and end-labeled by radioactive ATP using T4 polynucleotide kinase (NEB).

A total of 150 ng of double-stranded linear DNA was radiolabeled using the Klenow fragment (NEB) and $^{32}$P alpha dCTP in a 50 μl of reaction (50 mM Tris pH 6.8, 10 mM Magnesium acetate, 0.1 mM DTT, 0.05 mg/ml BSA, 0.6 mM dNTPs without dCTP) at room temperature for 60 mins. The radiolabeled probe was purified by 3 ml Sephadex G-50 column and precipitated overnight in 0.2 M sodium acetate pH 5.5/Ethanol followed by washes with 70% Ethanol and resuspension in 50 μl of ddH2O.

### $K_d$ Calculation
Densitometry data from various EMSA gels were obtained from ImageQuant (GE). Titration curves were generated by plotting protein concentration vs percentage shift of the radiolabeled probe. The $K_d$ value was analyzed by fitting a nonlinear regression curve with one site-specific binding mode in GraphPad Prism. *Tb*RAP1 fragments without any tag was used in EMSA for calculating $K_d$. The protein purity is >90% (Supplementary Fig. 3f).

### NMR spectroscopy
The concentrations of *Tb*RAP1$_{639-761}$ were 0.1 mM for the $^{15}$N-HSQC spectra, 0.6 mM for 2D $^{1}$H-$^{1}$H-NOESY and 1.0 mM for $^{15}$N-NOESY, HNCACB, CACB(CO)NH, $^{13}$C-HSQC and $^{13}$C-NOESY experiments. NMR samples were prepared in 100 mM phosphate buffer (Na$_2$HPO$_4$-NaH$_2$PO$_4$, pH 6.5) with 90% H$_2$O/10% D$_2$O or 99.9% D$_2$O. NMR spectra were acquired on Varian Inova 500, 750, or 800 MHz spectrometers at 298 K. The data were processed using NMRPipe[42] and analyzed using Sparky[43] and CCPN[44]. Backbone and side-chain resonance assignment were achieved via the standard heteronuclear triple resonance correlation experiments using $^{15}$N, $^{13}$C-double labeled *Tb*RAP1$_{639-761}$. Interproton distance restraints were generated from 2D/3D NOESY experiments using a mixing time of 100 ms. Hydrogen bond restraints were generated base on the nuclear overhauser enhancement (NOE) patterns and derived from Talos[45]. Initial structure models were generated using CNSsolve[46] using interproton distance, dihedral angle, and hydrogen bond restraints. Final structure refinement was performed using Xplor-NIH 3.3[47] using an implicit solvent potential[48]. Ten best structures of *Tb*RAP1$_{639-761}$ without restraint violation were selected.

### NMR titration assay
$^{15}$N-HSQC spectra were acquired with 0.1 mM $^{15}$N-labeled protein samples in 20 mM sodium sulfate, 150 mM NaCl, 1 mM EDTA, and 1 mM DTT at pH 6.5. NMR titrations were performed by adding unlabeled concentrated RNA (1-5 mM) to $^{15}$N labeled protein (0.1 mM) gradually. NMR spectra were acquired on Varian Inova 800 MHz spectrometer at 293 K.

### RNA cross-linking immunoprecipitation (CLIP)
500 million cells suspended in 1 X TDB (5 mM KCl, 80 mM NaCl, 1 mM MgSO$_4$, 20 mM Na$_2$HPO$_4$, 2 mM NaH$_2$PO$_4$, 20 mM glucose) were UV crosslinked (800 mJ) in a UV Stratalinker 2400 (Stratagene). Cells were then harvested by centrifugation, resuspended in the IP buffer (10 mM Tris•Cl pH 8.0, 150 mM NaCl, 0.1% NP-40, 1 X Protease inhibitor

cocktail (Roche), 40 units RNaseIn, 1% TritonX-100, 0.1% SDS, 100 μM TLCK, 1 μM Pepstatin A), and incubated on ice for 30 mins. Samples were centrifuged at 15,800 x *g* and 4 °C for 15 mins and the supernatant is collected as the lysate. 10% of the lysate was saved as input. The rest lysate was equally divided into two fractions, each incubating with 1.2 μg of monoclonal HA antibody 12CA5 (MSKCC Antibody & Bioresource Core)/*Tb*RAP1 rabbit antibody[15,17,18] or IgG conjugated with Dynabeads Protein-G (ThermoFisher) at 4 °C for 3 h with rotation. The IPed products were washed with wash buffer (10 mM Tris•Cl pH 8.0, 120 mM NaCl, 0.2% NP-40, 1% TritonX-100, 0.1% SDS) three times followed by washing with 1 X PBS once. After washing, the IP products were treated with proteinase K (200 μg) for 30 mins at 50 °C shaking at 450 rpm in a Thermomixer. RNA was then isolated from the IPed products using RNA STAT-60 (Tel-Test, Inc.) followed by DNase treatment and RNA purification over an RNeasy column (Qiagen). Reverse transcription was done using MMLV (Promega) according to the manufacturer's protocol followed by quantitative PCR using primers specific to various genes (Supplementary Table 4) and SsoAdvanced™ Universal SYBR® Green Supermix (Bio-Rad).

## Determination of *VSG2* mRNA stability

*TbRAP1*[F/mut] cells with and without the Cre expression (induced by adding 100 ng/ml Doxycycline for 29 h) were treated with 10 μg/ml Actinomycin D (Sigma) for 0, 15, 30, 45, 90, 120, or 150 mins. 40 million cells were harvested at each time point for isolation of total RNA using RNA STAT-60 (Tel-Test, Inc.). RNA samples were treated with DNase and purified on an RNeasy column (Qiagen). Quantitative RT-PCR was done the same way as described above. Data were processed using MS Excel and Graphpad Prism.

## Immunofluorescence (IF) analysis

Cells were fixed with 2% formaldehyde at RT for 10 min, permeabilized in 0.2% NP-40/1 X PBS at RT for 8 min, blocked by 1 X PBS/0.2% cold fish gelatin/0.5% BSA at RT twice, each for 10 min, followed by incubation with the primary antibody (12CA5 was diluted 1:2 K; *Tb*TRF, *Tb*RAP1, VSG6 rabbit antibodies and VSG3 monoclonal antibody were diluted 1:1 K; *Tb*TRF chicken antibody was diluted 1:200) at RT for 2 h and the secondary antibody at RT for 1 h. Cells were then washed with 1 X PBS/ 0.2% cold fish gelatin/0.5% BSA and 1 X PBS followed by staining with 0.5 μg/ml DAPI before mounting coverslips on slides. Images were taken by a DeltaVision Elite deconvolution microscope. Images were deconvolved using SoftWoRx.

## VSG switching assay

*TbRAP1*[+/+] and *TbRAP1*[/2FL] cells were first cultured for ~10.5 population doublings. At the end of culturing, 30 million cells were incubated with 10 μg of VSG2 monoclonal Antibody (IgM, MSKCC Antibody & Bioresource Core) on ice for 15 min. After washing 3 times with growth medium, cells were incubated with MACS beads conjugated with a rat anti-mouse IgM antibody (Miltenyi) on ice for 15 min followed by washing with growth medium twice. The mixture was then loaded onto an LD column, and cells in the flow-through fraction were collected and plated on 96-well dishes. 1/6 of the collected cells (equivalent to 5 millions of initial cell population) were evenly distributed onto three 96-well dishes. Similarly, 1/3 (equivalent to 10 millions of initial cell population) and 1/2 (equivalent to 15 millions of initial cell population) of the collected cells were evenly distributed into six and eight 96-well dishes, respectively. All recovered colonies were tested again by western slot blot using a VSG2 rabbit antibody (without the cross-reaction portion, 1:10,000), and VSG2-positive clones were excluded from switchers. Raw switching frequency was calculated by dividing the number of true switcher colonies by the initial cell number. To determine plating efficiency, cells were plated at 1 cell/well concentration onto 3 X 96-well plates. Plating efficiency was calculated by dividing the number of colonies grown up by 288. The final switching rate was calculated by normalizing raw switching frequency with plating efficiency and divided by the number of population doublings. Data were processed using MS Excel and Graphpad Prism.

## RNAseq

The Cre expression was induced by adding doxycycline in *TbRAP1*[F/2FQ] and *TbRAP1*[F/2FA&5A] cells for 30 h before total RNA was isolated and purified through RNeasy columns (Qiagen). All RNA samples were run on a BioAnalyzer 2100 (Agilent Technologies) using the Agilent RNA 6000 nano kit to verify the RNA quality before sent to Novogene for library preparation and RNA high throughput sequencing, which was the same as described in[18].

## RNAseq Data Analysis

RNAseq data were analyzed by Novogene:

**Quality control.** Raw reads of fastq format were first processed through Novogene perl scripts. In this step, clean reads were obtained by removing reads containing adapters, reads containing poly-N and low-quality reads. At the same time, Q20, Q30 and GC content of the clean reads were calculated. All downstream analyses were based on the clean reads with high quality.

**Reads mapping to the reference genome.** The *T. brucei* lister 427 genome TriTrypDB-45_TbruceiLister427_2018_Genome.fasta and its annotation TriTrypDB-45_TbruceiLister427_2018.gff were downloaded from the TriTryp DB and used as reference. Index of the reference genome was built using hisat2 2.1.0 and paired-end clean reads was aligned to the reference genome using HISAT2.

**Quantification of gene expression level.** HTSeq v0.6.1 was used to count the read numbers mapped to each gene. FPKM of each gene was calculated based on the length of the gene and the reads count mapped to this gene.

**Differential expression analysis.** Differential expression analysis of two conditions/groups (three biological replicates per condition) was performed using the DESeq R package (1.18.0). DESeq provides statistical routines for determining differential expression in digital gene expression data using a model based on the negative binomial distribution. The resulting *P*-values were adjusted using the Benjamini and Hochberg's approach for controlling the false discovery rate. Genes with an adjusted P-value <0.05 found by DESeq were assigned as differentially expressed.

## Reporting summary

Further information on research design is available in the Nature Portfolio Reporting Summary linked to this article.

## Data availability

The coordinates of the *Tb*RAP1 RRM structures generated in this study have been deposited in the Protein Data Bank (PDB) with the PDB identifier 7XRW and Biological Magnetic Resonance Data Bank (BMRB) with the identifier 36489. The NMR titration data generated in this study have been deposited at BMRB with the identifier 30936. The RNAseq data generated in this study have been deposited in NCBI's Gene Expression Omnibus and are accessible through GEO Series accession number GSE193394. Source data are provided with this paper. Reagents generated in this study are available upon request from the corresponding authors. Source data are provided with this paper.

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

## Acknowledgements

We thank Dr. Donny Licatolasi, Dr. Anton Komar, Dr. Kurt Runge, and Catherine Z. Wang for their comments on the manuscript. This work is supported by an NIH R01 grant AI066095 (Li), an NIH S10 grant S10OD025252 (Li), Research Grants Council grants PolyU 151062/18 M, 15103819, 15106421, R5050-18 and AoE/M-09/12 (Zhao), Shenzhen Basic Research Programs of China JCYJ20170818104619974 & JCYJ20210324133803009 (Zhao). Shenzhen Basic Research Program of China JCYJ20220818100215033 (Zhang). Research Grants Council grant C4041-18E (Wong, Zhang, Zhao). The publication cost is partly supported by GRHD at CSU and by PolyU.

## Author contributions

Study conception and design: B.L. and Y.Z.; Data collection: A.K.G., M.A., X.Y., A.S., S.A.S., X.P., Z.J., and B.L.; Data analysis and interpretation of results: B.L., Y.Z., K.B.W., and M.Z.; Draft manuscript preparation: B.L. and Y.Z.; Funding acquisition: B.L., Y.Z., K.B.W., and M.Z.; Supervision: B.L., Y.Z., and M.Z. All authors reviewed the results and approved the final version of the manuscript.

## Competing interests

The authors declare no competing interests.
