## [Peer Review File · Nature Communications]

The RRM-mediated RNA binding activity in *T. brucei* RAP1 is essential for VSG monoallelic expressionREVIEWER COMMENTS

Reviewer #1 (Remarks to the Author):

In this manuscript the authors present both experimental and structural evidence that the telomeric protein RAP1 of *Trypanosoma brucei* contains an RNA Recognition Motif (RRM) – a novel finding for RAP1 proteins. Evidence is also presented for the binding of TbRAP1 to consensus sequences of VSG 3'UTRs in vitro and to active VSG RNA in vivo. Using mutational analyses of key RNA binding residues for RNA binding and competition DNA binding experiments the authors show that this decreases the active VSG RNA level and derepresses silent VSGs. This suggests a mechanism by which monoallelic gene expression is regulated by abundant RNA transcripts from the active VSG that antagonizes the binding of Tb RAP1 to dsDNA to prevent repression. Overall a novel and interesting observation.

1) Overall the manuscript is very densely written so is difficult to follow. Each experimental section could do with an introduction as to why and how experiments were done. Some of the figures could be much better organised.

2) The RNA and DNA binding experiments could be much cleaner. A better definition of the DNA and RNA binding sites is required: How was the DNA – binding domain defined? Does it have any structure or is it just a bunch of basic residues from the NLS that contribute to binding? Is there any sequence specificity? What is the evidence that

Tb RAP1 binds directly to telomeric repeat and not via TRF as in mammalian cells?

Also for the RNA binding –which sequence does RAP1 recognize? A definition of the actual binding site within the RNA transcript would produce much cleaner results (Fig. 3). In particular the competition experiments in Fig. 7 could be more convincing.

Additionally the in vitro binding experiments need a more scientific presentation. RNA and DNA concentrations as well as protein concentrations should be given. Giving ngs is near meaning-less.

3) Abstract: It is a pity that there is no mention of the NMR structural analyses which is the strongest part of the manuscript. The last sentence of the abstract does not make sense.

Reviewer #2 (Remarks to the Author):

The manuscript reports on a novel RNA-mediated pathway for the regulation of the monoallelic expression of the Variant Surface Glycoprotein(s) (VSGs) of trypanosoma, a surface protein important for the parasite evasion of the host immune response. The VSGs loci are positioned in the proximity of telomeres and the authors have previously shown that the protein Rap1 binds to telomeric DNA and represses VSGs. They have also shown that this activity depends on a short R/K sequence in the carboxy-terminus of the protein.

Here, they report that the sequence amino-terminal to this R/K patch folds as RRM domain that binds specifically to RNA sequences in the 3'UTR of VSGs in vitro and in the cell. Then, using a range of construct and point mutations, they show this interaction regulates VSG abundance. Interestingly, the RRM and the R/K regions seem to play a different role in regulation, with the RRM being the key motif for the upregulation of the transcribed site, and the R/K being more important in the down-regulation of the repressed loci. The authors propose this functional difference stems from their different nucleic acid binding activity and propose a model for regulation.

This is an interesting paper that provide important molecular insight into a complex regulatory mechanism. The structural data are clear-cut and a different role for the two RNA-binding region is clearly shown via a range of RNA binding assays. This preliminary understanding allows to design a

well-thought range of mutants to explore the functional relevance of the two domain/types of binding, and the differences are, again, clear-cut. Conclusion are, in general, not overstated, although there is some doubt on the final model.

There are a few issues that need to be addressed prior to publication:

(1) The molecular understanding of RNA binding is somewhat preliminary, This is a molecular paper and protein-RNA recognition is at the core of it. It requires a clear model that explains how the RRM and R/K domains cooperate to recognize a specific sequence and how specific recognition is achieved.

The analysis of the NMR data on protein-nucleic acid interaction is not exhaustive and we do not have a clear model for how the R/K domain contributes to RNA binding. The authors should plot chemical shift changes against the protein sequence for the different protein-RNA titrations. This may help understanding some apparent discrepancies. For example, and importantly, when discussing the data in Figure 2A the authors mention that the resonances of residues C-terminal to the RRM do not shift upon RNA binding. However, looking at the comparison of short and long NA-binding constructs in panels 2b and 2e it is clear that the addition of the C-terminal amino acids, which contain the R/K, increases the affinity very substantially – and this is also mentioned by the authors. The NMR 15N-correlation spectra is fully assigned and it would be interesting to know whether the R/K residues are visible, and, if yes, whether they do shift. Although the assignment of the spectrum is provided, it is difficult for the reader to validate this. The author should do it and report their conclusions. Also important, the authors could map the binding of the RNAs on the protein structure, to see whether the same resonances of the RRM are affected in the presence and absence of the R/K motif and, in general whether the two 'domains' come together when binding RNA, or any other structural changes that may be highlighted by a full map.

In addition, the results from some of the experiments, for example the protein binding to the larger 3'UTR construct in the absence of the specific target sequence are also confusing. This could be the consequence of the lack of more quantitative data – in several instances what seem to be differences of a few fold are reported as binding/non binding and this make more difficult to rationalize the data in a quantitative model.

It could also be the consequence of the use very long oligos (>100 nts). It is sometimes difficult to define specificity in such long oligos, because of non-specific binding to a large number of overlapped non-specific sites. This would explain some of the EMSA data. I would suggest that using a minimal oligo would be useful to define specific binding and domain(s) contribution. I do understand the authors want to validate their data in larger targets, but there is no guarantee that any such large oligos would fold as it does in the cell. I would suggest that an oligo of 16 nt, such as the sequence underlined in the 34-nt long RNAs in Table S1, could accommodate binding of the two domains, with some nucleotides to spare. Individual RRM domain binds 2-8 nucleotides (4-6 in canonical binding as the one we seem to have here) and the R/K sequence is very short. The authors could attempt to validate their targets in a larger RNAs once they have a clear model of the interaction.

In summary, the authors could i) record NMR experiments on the RRM and RRM+R/K, wild type and with the different mutant, free and when bound to a minimal RNA construct that maintains the specificity of 34-VSG-3'UTR and analyze these data in depth as discussed above ii) quantify affinity in the same protein-RNA interactions using a biophysical technique while running in parallel EMSA assays which would report readily on stoichiometry. Once the contributions of the two 'domains' of the protein are quantified on a short construct one could try incrementally longer ones. Hopefully these data will provide a more conclusive model of how the two RNA binding regions participate to recognition.

Minor issues:

-In page 6, the authors mention the RRM is absent from RAP1 of higher eukaryotes. The authors should explain/show the evidence.

-In terms of the model, if the R/K contributes to RNA recognition it seems puzzling that, in Figure S4 and previous data there is essentially no effect on VSG2. Is it because the effect is too small to be detected this way? That seems possible, considering the effect of deleting the RRM is only 30%, but it should be discussed. Here a quantitative (and possibly even structural) model of how the two domains cooperate may be useful.

-In Figure S1C, colour coding should be used that allows to verify the quality of the rest of the alignment.

-In Figure S4c, the FL mutant protein is decreased by cre less than the wild type and the other mutants. While it seems logical that substituting an aromatic ring with a long hydrophobic chain has a weaker effect than substituting it with an Alanine, for example, the effect of this should be discussed.

Reviewer #3 (Remarks to the Author):

Review of RAP1 paper submitted to Nature communications.

This manuscript contains a first-rate, beautifully constructed and explained study that reveals a most interesting finding; namely that the TbRAP1 plays a key role in VSG monoallelic expression. Presentation of experiments is logical, and the extent and clarity of the experimental methods used, and data obtained, show the highest level of rigor to support the conclusions. Every experiment contains proper controls and all statements at the end of each subsection in the results ensures that the reader will easily follow along and grasp the important findings that from the experiments described above.

Some specific points-

The introduction is well constructed and nicely lays out the relevant background. The issue of where the interaction between the active VSG and TbRAP1 comes to mind as I read the introduction. Does this interaction support a role for TbRAP1 in the cytoplasm? This question remains an open one until the very end of the manuscript. Some mention early on of TbRAP1 only being in the nucleus would eliminate this bothersome question as one reads the paper.

The finding that the RNA and DNA binding of RAP1 are coordinated is extremely interesting and is a very important addition to the growing literature that shows 'new' RNA binding protein function in cellular nuclei. A review of nucleic acid binding proteins by 'traditional' RNA or DNA binding proteins could be a well-received and timely follow up paper by this group.

Very nice to see SNAP 50 used as a control. The more SNAP50 is included in studies the better.

Line 126 should say fig 1d, not 1c. line 129 should say fig 1c, not 1d.

The presence of RNP1 and 2 within RRM is often hard to clarify. This seems to be the case here.

Maybe adding the RNP1 and 2 to an enlarged version of the diagram of the protein in fig 1c would really help. This enlarged, and more detailed showing of regions ~639 to 761 would really help the reader follow along the text that discusses RNP1 and 2, as well as the phenylalanine mutations, etc, that are important in the study.

Line 179 should list RNP1 before RNP2.

In figure 3b, is the conclusion that more than one molecule of protein binds the RNA based on the two closely spaced gel shift bands?

The finding that RAP1 bound a 170 nt RNA without consensus VSG 3'UTR sequences is clearly shown, and the description of the authors' interpretation is excellent and reveals a clarity of thinking that is truly admirable.

AS noted above, a detailed drawing of the amino acids in the 639-761 region would help the reader follow along in several sections. I refer here to the section from lines 237-246. These excellent data would be better appreciated if one could refer back to an expanded sequence included as an additional panel in fig 1 (or as a new figure?)

Including estimated binding, K_d values, in the project really supports the arguments for a model based on competitive binding. Excellent model developed and nicely supported by the data. In the excellent discussion, the problem of defining exact RNA recognition sequences used by RNA binding proteins is brought up. This point is nicely made here; more scientists need to realize that RNA binding proteins have many traits that are different from DNA binding proteins.

Responses to Reviewers' comments

We appreciate the careful and thorough evaluation of our manuscript by the three reviewers. We are grateful for the constructive suggestions and comments.

As suggested by the reviewers, we have now performed the following experiments:

- (1) We did NMR titration studies of TbRAP1-RRM+DB and TbRAP1-RRM using a shorter RNA substrate (16-VSG-UTR);
- (2) We plotted NMR chemical shifts for TbRAP1-RRM+DB and TbRAP1-RRM after titration with 34-VSG-UTR and 16-VSG-UTR;
- (3) We performed fluorescent polarization experiments and estimated the binding affinity (K_d) of TbRAP1-RRM+DB, TbRAP1-RRM, and TbRAP1-RRM+DB-5A to 16-VSG-UTR;
- (4) We performed additional EMSA to confirm TbRAP1-RRM+DB's binding on 16-VSG-UTR;
- (5) We performed RNA CLIP assay in TbRAP1 Δ DB and TbRAP1-5A mutants;
- (6) We characterized the mutant phenotypes in the TbRAP1-2FA mutant;
- (7) We performed more qRT-PCR analysis in the TbRAP1-2FA&5A mutant to examine the active VSG RNA level changes.

The new data are shown in the following new figures:

- (1) Inset of Fig. 1c; (2) Fig. 2b, e; (3) Fig. 3j; (4) TbRAP1 Δ DB and TbRAP1-5A data in Fig. 4a; (5) New data from TbRAP1-2FA&5A in Fig. 5f, g; (6) Supplementary Fig S1e; (7) Supplementary Fig S2c-j; (8) Supplementary Fig S4f, k, m; and (9) Supplementary Fig S6c.

We have addressed the reviewers' critiques point-by-point, which is shown below. The Reviewer's critiques are in the Palatino font and italicized. Our responses are in the normal Arial font.

We truly hope that we have addressed all reviewers' critiques satisfactorily.

Reviewer #1 (Remarks to the Author):

In this manuscript the authors present both experimental and structural evidence that the telomeric protein RAP1 of Trypanosoma brucei contains an RNA Recognition Motif (RRM) – a novel finding for RAP1 proteins. Evidence is also presented for the binding of TbRAP1 to consensus sequences of VSG 3'UTRs in vitro and to active VSG RNA in vivo. Using mutational analyses of key RNA binding residues for RNA binding and competition DNA binding experiments the authors show that this decreases the active VSG RNA level and derepresses silent VSGs. This suggests a mechanism by which monoallelic gene expression is regulated by abundant RNA transcripts from the active VSG that antagonizes the binding of Tb RAP1 to dsDNA to prevent repression. Overall a novel and interesting observation.

1) Overall the manuscript is very densely written so is difficult to follow. Each experimental section could do with an introduction as to why and how experiments were done. Some of the figures could be much better organised.

We thank the reviewer's comment. Indeed, a large amount of data has been included in the manuscript to demonstrate the vigor of our study and to validate our novel finding. We have followed this reviewer's advice and added brief introduction in several experimental section to explain the rationale of our experiments so that readers can understand our data and the conclusion.

2) The RNA and DNA binding experiments could be much cleaner. A better definition of the DNA and RNA binding sites is required: How was the DNA – binding domain defined? Does it have any structure or is it just a bunch of basic residues from the NLS that contribute to binding? Is there any sequence specificity? What is the evidence that Tb RAP1 binds directly to telomeric repeat and not via TRF as in mammalian cells?

We thank the reviewer for these questions. We have reported our work on TbRAP1's DB domain, its DNA binding activities and their functions in Afrin et al. 2020. *Sci. Adv.* **6**, eabc406. We have also reported our work on TbRAP1's NLS region in Afrin et al. 2020. *mSphere* **5**, e00027-20.

Briefly, the DNA binding domain (DB) was defined as the aa 734 to 761 region and the NLS is the aa 727-741 region. The ⁷³⁷RKRRR₇₄₁ patch is essential for both the DB domain and the NLS region. TbRAP1 has both ssDNA and dsDNA binding activities mediated by the ⁷³⁷RKRRR₇₄₁ patch, which are electrostatics based and sequence non-specific.

The NMR structure reported in this manuscript shows that the DB domain, including the ⁷³⁷RKRRR₇₄₁ patch, forms a long and flexible loop C-terminal to the RRM region. We have revised the manuscript to clarify about the definition of the DB domain (page 3; line 85) and to highlight the flexible loop structure of the DB domain in Supplementary Fig. S1b (pages 5-6; lines 138-140).

In terms of whether *TbRAP1* binds to telomeric repeats directly or via *TbTRF*, our previous study (Afrin et al. 2020. *Sci. Adv.* **6**, eabc406) showed that *TbRAP1* was still associated with the telomere chromatin in *TbTRF*-depleted cells but mutation of ⁷³⁷RKRRR₇₄₁ to ⁷³⁷AAAAA₇₄₁ disrupted the association of *TbRAP1* with the telomere chromatin *in vivo*. These results suggest that, in *T. brucei*, targeting *TbRAP1* to the telomere relies more critically on its DB domain than *TbTRF*.

Also for the RNA binding –which sequence does RAP1 recognize? A definition of the actual binding site within the RNA transcript would produce much cleaner results (Fig. 3).

In the revised manuscript, we have now shown that *TbRAP1* RRM can recognize the 16-mer consensus sequence of VSG 3'UTRs (Fig. 3j; Supplementary Fig 2c-j). However, we have also stated that it is very likely that *TbRAP1* RRM may recognize additional RNA sequences, a common feature for the RRM fold (page 9; lines 252-255; page 15, lines 424-430). Future omics studies will be necessary to identify all *TbRAP1* RRM RNA substrates.

In particular the competition experiments in Fig. 7 could be more convincing.

In our previous study (Afrin et al. 2020. *Sci. Adv.* **6**, eabc406), we have shown that *TbRAP1* binds longer DNA molecules with higher affinity, and it does not bind dsDNA shorter than 60 bp. Therefore, it is expected that *TbRAP1* will bind shorter RNA substrate (such as 35-VSG-UTR) much better than a 35 bp dsDNA substrate. Therefore, we have not performed more DNA/RNA binding competition assays using shorter substrates. In addition, we have stated "To investigate whether such competition applies to shorter DNA or RNA substrates, we further compared *TbRAP1*₆₃₉₋₇₆₁ binding on 80-dsDNA and 81-VSG-UTR (Supplementary Table ST2), as the shortest ssDNA and dsDNA that *TbRAP1* can bind is ~60 nt and 60 bp, respectively ¹⁸." in the revised manuscript (Page 14, lines 401-404).

Additionally the in vitro binding experiments need a more scientific presentation. RNA and DNA concentrations as well as protein concentrations should be given. Giving ngs is near meaning-less.

We thank the reviewer for this suggestion. We have indicated all protein concentrations in EMSA analyses. The RNA and DNA concentrations are in general the same for EMSA experiments, which are indicated in the "EMSA" paragraph of the materials and methods section.

3) Abstract: It is a pity that there is no mention of the NMR structural analyses which is the strongest part of the manuscript. The last sentence of the abstract does not make sense.

We thank the reviewer for this comment. We have now revised the abstract and specifically mentioned that our NMR structural analyses identify the *TbRAP1* RRM domain. We have also revised the Abstract in general.

Reviewer #2 (Remarks to the Author):

The manuscript reports on a novel RNA-mediated pathway for the regulation of the monoallelic expression of the Variant Surface Glycoprotein(s) (VSGs) of trypanosoma, a surface protein important for the parasite evasion of the host immune response. The VSGs loci are positioned in the proximity of telomeres and the authors have previously shown that the protein Rap1 binds to telomeric DNA and represses VSGs. They have also shown that this activity depends on a short R/K sequence in the carboxy-terminus of the protein.

Here, they report that the sequence amino-terminal to this R/K patch folds as RRM domain that binds specifically to RNA sequences in the 3'UTR of VSGs in vitro and in the cell. Then, using a range of construct and point mutations, they show this interaction regulates VSG abundance. Interestingly, the RRM and the R/K regions seem to play a different role in regulation, with the RRM being the key motif for the upregulation of the transcribed site, and the R/K being more important in the down-regulation of the repressed loci. The authors propose this functional difference stems from their different nucleic acid binding activity and propose a model for regulation.

This is an interesting paper that provide important molecular insight into a complex regulatory mechanism. The structural data are clear-cut and a different role for the two RNA-binding region is clearly shown via a range of RNA binding assays. This preliminary understanding allows to design a well-thought range of mutants to explore the functional relevance of the two domain/types of binding, and the differences are, again, clear-cut. Conclusion are, in general, not overstated, although there is some doubt on the final model.

We thank the reviewer for this positive evaluation of our manuscript.

There are a few issues that need to be addressed prior to publication:

(1) The molecular understanding of RNA binding is somewhat preliminary, This is a molecular paper and protein-RNA recognition is at the core of it. It requires a clear model that explains how the RRM and R/K domains cooperate to recognize a specific sequence and how specific recognition is achieved.

We thank the reviewer for this comment, and we agree that how the RRM region and the DB domain cooperate for effective RNA binding is the core question to be addressed.

In the revised manuscript, we have added the following two lines of data to define the molecular determinants of the RNA binding activity.

First, using NMR titration, we examined the binding of TbRAP1-RRM+DB and TbRAP1-RRM on both 34-VSG-UTR and 16-VSG-UTR, the latter of which contains only the 16-

mer consensus sequence within the 34-VSG-UTR oligo (Fig. 2a, d; Supplementary Fig. S2c, e). We plotted the chemical shifts induced by the RNA substrate over the entire protein sequence for these NMR titration experiments (Fig. 2b, e; Supplementary Fig. S2d, f). These plots confirm that, while RRM-only can bind to the 3'UTR region of the VSG RNA with weak affinity, this binding is significantly strengthened by DB.

Second, we used the fluorescence polarization assay as a biophysical technique to assess the RNA binding activity of *TbRAP1*. Fluorophore-labeled 16-VSG-UTR was titrated to *TbRAP1*-RRM+DB, *TbRAP1*-RRM and *TbRAP1*-RRM+DB-5A (Supplementary Fig S2g-i). These data confirm the NMR titration results that cooperation between RRM and DB leads to stronger RNA binding.

In terms of sequence specificity of this RNA binding activity, we compared the NMR chemical shifts induced by 34-VSG-UTR and 16-VSG-UTR. For *TbRAP1*-RRM, the chemical shifts induced by these two RNA substrates were similar (Fig. 2e vs. Supplementary Fig S2f), suggesting that the 16-nt oligo is sufficient for RNA recognition by *TbRAP1*. For *TbRAP1*-RRM+DB, 34-VSG-UTR induced much stronger chemical shifts than 16-VSG-UTR (Fig. 2b vs. Supplementary Fig S2d), implying that the DB domain may contact nucleotides flanking the 16-mer consensus sequence of VSG 3'UTRs in the 34-VSG-UTR in a sequence non-specific manner. Similar results were observed in our EMSA analysis, where *TbRAP1*-RRM+DB showed a stronger affinity to 34-VSG-UTR than 16-VSG-UTR oligo (Supplementary Fig S2j). Considering that RRM is known to bind multiple RNA sequences, we believe future studies such as HITS-CLIP are needed to fully delineate the RNA sequences that can be recognized by *TbRAP1*.

In summary, our data indicate that *TbRAP1*-RRM alone binds to the 16-mer consensus sequence of VSG 3'UTRs with a moderate affinity. This binding is significantly strengthened by the DB domain, which binds to both DNA and RNA in electrostatics-based and sequence-non-specific manner.

We have revised the manuscript to reflect these new results.

The analysis of the NMR data on protein-nucleic acid interaction is not exhaustive and we do not have a clear model for how the R/K domain contributes to RNA binding. The authors should plot chemical shift changes against the protein sequence for the different protein-RNA titrations. This may help understanding some apparent discrepancies. For example, and importantly, when discussing the data in Figure 2A the authors mention that the resonances of residues C-terminal to the RRM do not shift upon RNA binding. However, looking at the comparison of short and long NA-binding constructs in panels 2b and 2e it is clear that the addition of the C-terminal amino acids, which contain the R/K, increases the affinity very substantially – and this is also mentioned by the authors. The NMR ¹⁵N-correlation spectra is fully assigned and it would be interesting to know whether the R/K residues are visible, and, if yes, whether they do shift. Although the assignment of the spectrum is provided, it is difficult for the reader to validate this. The author should do it and report their conclusions. Also important, the authors could map the binding of the RNAs on the protein structure, to see whether the same resonances of the RRM are affected in the presence and absence of the R/K motif and, in general whether the two

'domains' come together when binding RNA, or any other structural changes that may be highlighted by a full map.

We are grateful for the reviewer's suggestion. As explained in our reply to the last comment, we have conducted more thorough analyses by plotting the chemical shifts induced by RNA substrates over the entire protein sequence of *TbRAP1*-RRM+DB, *TbRAP1*-RRM, using both 34-VSG-UTR (Fig. 2b, e) and 16-VSG-UTR (Supplementary Fig S2d, f).

As the reviewer pointed out, upon such closer analysis, it is indeed clear that residues in the DB domain also showed noticeable chemical shifts, although smaller than those for the RNP1 and RNP2 motifs in RRM (Fig. 2b; Supplementary Fig S2d). Thus, the DB domain, which binds to both DNA and RNA in electrostatics-based and sequence-non-specific manner, indeed cooperate with RRM for a stronger RNA binding.

In terms of how the RRM and DB domains cooperate for RNA binding, our NMR structure of *TbRAP1*-RRM+DB shows that the DB domain is a highly flexible loop and there is little inter-domain interactions between RRM and DB regions in the absence of the RNA substrate (Supplementary Fig S1b). Additionally, the NMR chemical plots reveal that RNA perturbed the same set of residues in the RRM region, whether the DB domain was present or absent (Fig. 2b vs. 2e for 34-VSG-UTR and Supplementary Fig S2d vs. S2f for 16-VSG-UTR). These observations suggest that the flexible DB domain does not have noticeable interaction with the RRM domain even in the presence of a bound RNA substrate. It is possible that the RRM and DB domains interact with different parts of the RNA substrate independently and their cooperativity is due to this bi-valent binding mode. We have added a paragraph of discussion on this (page 15; lines 433-441).

In addition, the results from some of the experiments, for example the protein binding to the larger 3'UTR construct in the absence of the specific target sequence are also confusing. This could be the consequence of the lack of more quantitative data – in several instances what seem to be differences of a few fold are reported as binding/non binding and this make more difficult to rationalize the data in a quantitative model.

It could also be the consequence of the use very long oligos (>100 nts). It is sometimes difficult to define specificity in such long oligos, because of non-specific binding to a large number of overlapped non-specific sites. This would explain some of the EMSA data. I would suggest that using a minimal oligo would be useful to define specific binding and domain(s) contribution. I do understand the authors want to validate their data in larger targets, but there is no guarantee that any such large oligos would fold as it does in the cell. I would suggest that an oligo of 16 nt, such as the sequence underlined in the 34-nt long RNAs in Table S1, could accommodate binding of the two domains, with some nucleotides to spare. Individual RRM domain binds 2-8 nucleotides (4-6 in canonical binding as the one we seem to have here) and the R/K sequence is very short. The authors could attempt to validate their targets in a larger RNAs once they have a clear model of the interaction.

We are grateful for reviewer's suggestion. We agree that using long RNA substrates cannot provide clear understanding about *TbRAP1* RRM's binding specificity. Using NMR titration, fluorescent polarization, and EMSA, we have now tested *TbRAP1*-RRM+DB and *TbRAP1*-RRM's binding on 16-VSG-UTR that contains only the 16-mer consensus sequence of VSG 3'UTRs. *TbRAP1*-RRM can clearly bind 16-VSG-UTR in both NMR titration and fluorescent polarization, and *TbRAP1*-RRM+DB can bind 16-VSG-UTR in all three *in vitro* assays with a higher affinity than *TbRAP1*-RRM.

Since individual RRM domain binds 2-8 nucleotides and many RRM domains have been reported to recognize more than one RNA substrates, we anticipate that *TbRAP1*-RRM may recognize more than one RNA sequence as well, which is consistent with the observation that *TbRAP1*-RRM binds longer RNA substrates in a seemingly sequence-non-specific manner. We have now stated this possibility in the revised manuscript (page 9, lines 252-255; page 15, lines 424-430).

In summary, the authors could i) record NMR experiments on the RRM and RRM+R/K, wild type and with the different mutant, free and when bound to a minimal RNA construct that maintains the specificity of 34-VSG-3'UTR and analyze these data in depth as discussed above

We thank the reviewer for summarizing the suggested NMR studies and analysis. As explained above, we have performed the suggested experiments including (1) plotting the chemical shifts induced by RNA titration over the entire protein sequence of *TbRAP1*-RRM+DB and *TbRAP1*-RRM, using both 34-VSG-UTR (Fig. 2b, e) and 16-VSG-UTR (Supplementary Fig S2d, f); and (2) conducting the fluorescence polarization assay as a biophysical approach to assess the binding activity of *TbRAP1*-RRM+DB, *TbRAP1*-RRM, and *TbRAP1*-RRM+DB-5A for 16-VSG-UTR (Supplementary Fig S2g-i). The obtained results were analyzed accordingly.

ii) quantify affinity in the same protein-RNA interactions using a biophysical technique while running in parallel EMSA assays which would report readily on stoichiometry. Once the contributions of the two 'domains' of the protein are quantified on a short construct one could try incrementally longer ones. Hopefully these data will provide a more conclusive model of how the two RNA binding regions participate to recognition.

To provide more quantitative results, we have measured the binding affinities (estimated K_d values) for *TbRAP1*-RRM+DB and *TbRAP1*-RRM on different RNA substrates using both fluorescent polarization (Supplementary Fig S2g-i) and EMSA (Supplementary Fig S2j). We have also validated by NMR titration and fluorescent polarization that *TbRAP1*-RRM can recognize the 16-mer consensus sequence of VSG 3'UTRs, and this binding is enhanced by the DB domain. These NMR analysis and fluorescent polarization data corroborate with findings from our EMSA assays that were conducted with longer RNA substrates of 35-nt, 81-nt and 170-nt length.

Minor issues:

-In page 6, the authors mention the RRM is absent from RAP1 of higher eukaryotes. The authors should explain/show the evidence.

We thank the reviewer's suggestion. We have now done a sequence alignment of the C-terminal half of various RAP1 homologs (Fig. S1e). This strongly suggests that RAP1 homologs in higher eukaryotes do not have any RRM domain, although sequence alignment is limited and does not provide as much information as structural analyses.

-In terms of the model, if the R/K contributes to RNA recognition it seems puzzling that, in Figure S4 and previous data there is essentially no effect on VSG2. Is it because the effect is too small to be detected this way? That seems possible, considering the effect of deleting the RRM is only 30%, but it should be discussed. Here a quantitative (and possibly even structural) model of how the two domains cooperate may be useful.

We thank the reviewer for raising this critical point. To further investigate the R/K patch's role in RNA binding, we have now done many more *in vivo* experiments in addition to the NMR and fluorescent polarization data stated above.

First, we have done RNA CLIP to examine the *Tb*RAP1-VSG RNA interaction in *Tb*RAP1 Δ DB and *Tb*RAP1-5A mutants, both of which mutated the R/K patch. To our great surprise, *Tb*RAP1 Δ DB and *Tb*RAP1-5A do not bind the active VSG RNA *in vivo*, although both NMR titration and EMSA clearly showed that *Tb*RAP1-5A can bind RNA substrates containing VSG 3'UTR sequences. Since *Tb*RAP1's RNA binding activity is moderate in the absence of the DB domain and a high concentration of the VSG RNA appears to be important for the *Tb*RAP1-VSG RNA interaction, we speculate that loss of the DNA binding activity in *Tb*RAP1-5A and *Tb*RAP1 Δ DB removes *Tb*RAP1 from the telomere chromatin (as demonstrated previously in Afrin et al. 2020. *Sci. Adv.* **6**, eabc406) and prevents *Tb*RAP1 from getting access to a high concentration of the active VSG RNA at the active ES, thereby disrupting the *Tb*RAP1-VSG RNA interaction (page 10, lines 285-292).

Second, we only observed ~ 10% decrease in the active VSG RNA level in *Tb*RAP1 Δ DB and *Tb*RAP1-5A mutants, a milder phenotype than *Tb*RAP1 RRM point mutants. This result could be explained by the dual roles the DB domain plays in mediating the DNA and RNA binding activities. *Tb*RAP1 RRM point mutants have normal dsDNA binding activity and are still associated with the telomere chromatin (Fig. 4c, d; Fig. S4m). Therefore, we expect that the *Tb*RAP1-mediated VSG silencing is still intact in these mutants. This is why when *Tb*RAP1 does not interact with the active VSG RNA, the active VSG will be silenced by *Tb*RAP1, the same as other telomeric VSGs, and we observed a significant decrease in the active VSG RNA level. On the other hand, *Tb*RAP1 Δ DB and *Tb*RAP1-5A are no longer associated with the telomere chromatin (Afrin et al. 2020. *Sci. Adv.* **6**, eabc406), and the *Tb*RAP1-mediated VSG silencing is defective in these mutants. Therefore, the active VSG is not silenced by *Tb*RAP1 even if *Tb*RAP1 does not bind the active VSG RNA. This is why we did not observe a significant decrease in the active VSG RNA level in *Tb*RAP1 Δ DB and *Tb*RAP1-5A mutants.

Finally, our new data regarding the dual roles of the DB domain suggest that we should observe the same phenotype in the *Tb*RAP1-2FA&5A mutant as in *Tb*RAP1 Δ DB and *Tb*RAP1-5A mutants, as *Tb*RAP1-2FA&5A does not bind the active VSG RNA or associate with the telomere chromatin (Fig. 4a, b), the same as *Tb*RAP1 Δ DB and

TbRAP1-5A. We therefore repeated many more qRT-PCR analyses (two more biological repeats, each containing 3 technical repeats). We found that in fact the active VSG RNA level is only decreased ~13% in TbRAP1-2FA&5A, which is not significantly different from that in TbRAP1-5A.

In summary, we now found that the DNA binding activity mediated by the R/K patch in the DB domain is critical for the *in vivo* TbRAP1-VSG RNA interaction. Recruiting TbRAP1 to the telomere chromatin is likely a prerequisite for TbRAP1's binding to the active VSG RNA *in vivo*.

-In Figure S1C, colour coding should be used that allows to verify the quality of the rest of the alignment.

We thank the reviewer for this suggestion. We have revised supplementary Fig S1c and highlighted conserved residues.

-In Figure S4c, the FL mutant protein is decreased by cre less than the wild type and the other mutants. While it seems logical that substituting an aromatic ring with a long hydrophobic chain has a weaker effect than substituting it with an Alanine, for example, the effect of this should be discussed.

We thank the reviewer for this comment. Cre expression only removes the loxP-flanked WT TbRAP1 allele. In Fig. S5c, we observed that the WT TbRAP1 was completely depleted after Cre induction (bottom western where the mutant and the WT TbRAP1 proteins were separated with a long electrophoresis). The expression of the TbRAP1-2FL mutant is not expected to change upon Cre induction. We have now added a comment that “substituting an aromatic ring (in the phenylalanine residue) with a long hydrophobic chain (in the leucine residue) likely has a weaker effect than substituting it with an alanine” (page 11, lines 313-315).

Reviewer #3 (Remarks to the Author):

Review of RAP1 paper submitted to Nature communications.

This manuscript contains a first-rate, beautifully constructed and explained study that reveals a most interesting finding; namely that the TbRAP1 plays a key role in VSG monoallelic expression. Presentation of experiments is logical, and the extend and clarity of the experimental methods used , and data obtained, show the highest level of rigor to support the conclusions. Every experiment contains proper controls and all statements at the end of each subsection in the results ensures that the reader will easily follow along and grasp the important findings that from the experiments described above.

We thank the reviewer for this positive evaluation of our manuscript.

Some specific points-

The introduction is well constructed and nicely lays out the relevant background. The issue of where the interaction between the active VSG and TbRAP1 comes to mind as I read the introduction. Does this interaction support a role for TbRAP1 in the cytoplasm? This question remains an open one until the very end of the manuscript. Some mention early on of TbRAP1 only being in the nucleus would eliminate this bothersome question as one reads the paper.

We thank the reviewer for this suggestion. We have added in the introduction section that TbRAP1 is a nuclear protein (page 3, line 79).

The finding that the RNA and DNA binding of RAP1 are coordinated is extremely interesting and is a very important addition to the growing literature that shows 'new' RNA binding protein function in cellular nuclei. A review of nucleic acid binding proteins by 'traditional' RNA or DNA binding proteins could be a well-received and timely follow up paper by this group. Very nice to see SNAP 50 used as a control. The more SNAP50 is included in studies the better.

We thank the reviewer for these comments. We will definitely think of a follow-up review paper as suggested by this reviewer.

Line 126 should say fig 1d, not 1c. line 129 should say fig 1c, not 1d.

We realized that our description was not clear. We have now revised this part and referred to the correct figure panels.

The presence of RNP1 and 2 within RRM is often hard to clarify. This seems to be the case here. Maybe adding the RNP1 and 2 to an enlarged version of the diagram of the protein in fig 1c would really help. This enlarged, and more detailed showing of regions ~639 to 761 would really help the reader follow along the text that discusses RNP1 and 2, as well as the phenylalanine mutations, etc, that are important in the study.

We thank the reviewer for this suggestion. An enlarged image of the TbRAP1-MybLike domain has been added to Fig. 1c with a labeling of the positions of RNP1 and RNP2.

Line 179 should list RNP1 before RNP2.

We have revised this sentence and made it clear that F655 is in RNP2 while F694 is in RNP1 (page 6, line 145).

In figure 3b, is the conclusion that more than one molecule of protein binds the RNA based on the two closely spaced gel shift bands?

Yes.

The finding that RAP1 bound a 170 nt RNA without consensus VSG 3'UTR sequences is clearly shown, and the description of the authors' interpretation is excellent and reveals a clarity of thinking that is truly admirable.

We thank the reviewer for this comment.

As noted above, a detailed drawing of the amino acids in the 639-761 region would help the reader follow along in several sections. I refer here to the section from lines 237-246. These excellent data would be better appreciated if one could refer back to an expanded sequence included as an additional panel in fig 1 (or as a new figure?)

We have now added an image of the enlarged *Tb*RAP1-MybLike region in Fig. 1c.

Including estimated binding, K_d values, in the project really supports the arguments for a model based on competitive binding. Excellent model developed and nicely supported by the data.

We thank the reviewer for this comment.

In the excellent discussion, the problem of defining exact RNA recognition sequences used by RNA binding proteins is brought up. This point is nicely made here; more scientists need to realize that RNA binding proteins have many traits that are different from DNA binding proteins.

We thank the reviewer for this comment.

REVIEWER COMMENTS

Reviewer #2 (Remarks to the Author):

The authors have performed additional experiments and answered my comments. There is one point where I disagree with their interpretation of the data.

Their authors state the size of chemical shift changes in Figures 2e and S2f are very similar, while the size of chemical shift changes in Figures 2b and S2d, are quite different. As far as I can see, in both cases the difference is between 1.5 and 2 fold (~1.6 vs 1.9). If the changes are indeed so similar the implication is that the tail of the protein likely binds within the 16mer RNA. Could the authors revise and correct where required in the paper (results and conclusions). If the authors feel the small difference may be significant perhaps they can quantify and discuss in the paper.

A second minor point is that, the increase in affinity linked to the interaction of the tail with the RNA is of a few fold. The authors show such a difference is functionally relevant, which has been observed in a number of other complex combinatorial interactions. It would be useful to discuss briefly in the paper, and cite some other examples.

Reviewer #3 (Remarks to the Author):

This is a comprehensive revision that adds a new biophysical measurement method- fluorescent polarization- to the analysis of the protein-RNA interactions between TbRAP1 and the 3'UTR of VSG. The authors have responded in detail- in many cases with additional experiments- to all reviewers' insightful comments and have now produced an experimentally well supported, and intellectually well argued model for the bifunctional role of RAP1 in regulating VSG mRNA uniallelic expression and DNA binding. The use of mutants that include DB-minus protein is very informative as are the extensive NMR and EMSA assays.

Reviewer #4 (Remarks to the Author):

In this revised version of the manuscript, the authors have added new experimental data and modified the text to address reviewers' concerns. I agree with the original reviewers that the discovery of an RRM in TbRAP1 and the observation that mutually exclusive RNA and DNA binding contributes to the maintenance of VSG mono-allelic expression are interesting and novel findings. However, I also agree that the manuscript is difficult to follow, even in its revised version. This is not an inevitable consequence of a manuscript containing a large amount of data as asserted in the rebuttal. It is a question of sentence structure and wording. I highly recommend that the authors go over the manuscript together with a native English speaker to improve clarity throughout.

In terms of the points made by reviewer 1, the revisions ultimately do not resolve which RNA sequence is bound by TbRAP1 and together with the non-sequence specific interactions with DNA and the unexpected binding to 170-no-VSG leave the puzzle of specificity in MAE regulation unanswered. It is not entirely clear to this reviewer why the authors did not expand on binding experiments with a systematic panel of short RNA substrates that would have defined the specific sequence motif recognized by the RRM. Although the in vitro binding experiments are now presented in "a more scientific manner" by listing molar concentrations for substrate and protein, the method section makes it clear that the manuscript harbors a collection of binding assays carried out under different conditions. The reader is left wondering which condition applies to which assay. For example, whether EMSA assays are run through 0.8% agarose, 1.2% agarose, or 5% polyacrylamide can have profound effects on observed mobility shifts and apparent binding constants. Furthermore, variable levels of radioactive material appear to be retained in the wells in all EMSA panels, but the gels are cropped in

a manner that prevents further assessment of well-shifts. EMSA gels are not included in the source data raw data images.

In summary, the authors propose an interesting model in line with recent findings of how mutually exclusive DNA and RNA binding contributes to the regulation of transcription in a locus specific manner. Unfortunately, the manuscript lacks clarity, and the mechanism ultimately remains elusive.

Other comments:

Figure 1b, correct: CLIP/input

The normalization appears problematic as it artificially removes all variation among experiments for the IgG pulldown, thereby inflating significance of difference for RAP1 Ab.

The (UUAGGG)₄ probe used to assess TERRA binding is problematic as it folds into highly stable intramolecular G-quadruplex structures making the RNA unavailable for sequence-specific binding. Commonly 3.5 repeats are used for this reason.

It is surprising that 2FQ and 2FA mutants result in acute growth arrest, as expression of multiple VSGs would not be expected to cause this phenotype. The authors should comment on what is responsible for this acute phenotype.

REVIEWER COMMENTS

Reviewer #2 (Remarks to the Author):

The authors have performed additional experiments and answered my comments. There is one point where I disagree with their interpretation of the data.

Their authors state the size of chemical shift changes in Figures 2e and S2f are very similar, while the size of chemical shift changes in Figures 2b and S2d, are quite different. As far as I can see, in both cases the difference is between 1.5 and 2 fold (~1.6 vs 1.9). If the changes are indeed so similar the implication is that the tail of the protein likely binds within the 16mer RNA. Could the authors revise and correct where required in the paper (results and conclusions). If the authors feel the small difference may be significant perhaps they can quantify and discuss in the paper.

We thank the reviewer for this point. Our previous analysis was based solely on comparing the magnitude of chemical shifts, but we agree it is more meaningful to compare the fold of change. Indeed, calculating the fold of changes leads to the conclusion that for both 34-VSG-UTR and 16-VSG-UTR oligos, the DB domain plays a similar role in supporting effective RNA binding by the RRM domain. We have revised this section as suggested (Page 7, lines 204 – 209).

A second minor point is that, the increase in affinity linked to the interaction of the tail with the RNA is of a few fold. The authors show such a difference is functionally relevant, which is has been observed in a number of other complex combinatorial interactions. It would be useful to discuss briefly in the paper, and cite some other examples.

We thank the reviewer for this suggestion. We have added a section in the Discussion about this and cited other examples (Page 15-16, lines 449 – 462).

Reviewer #3 (Remarks to the Author):

This is a comprehensive revision that adds a new biophysical measurement method- fluorescent polarization- to the analysis of the protein-RNA interactions between TbRAP1 and the 3'UTR of VSG. The authors have responded in detail- in many cases with additional experiments- to all reviewers' insightful comments and have now produced an experimentally well supported, and intellectually well argued model for the bifunctional role of RAP1 in regulating VSG mRNA uniallelic expression and DNA binding. The use of mutants that include DB-minus protein is very informative as are the extensive NMR and EMSA assays.

We are grateful for the reviewer's positive comments on our revised manuscript.

Reviewer #4 (Remarks to the Author):

In this revised version of the manuscript, the authors have added new experimental data and modified the text to address reviewers' concerns. I agree with the original reviewers that the discovery of an RRM in

TbRAP1 and the observation that mutually exclusive RNA and DNA binding contributes to the maintenance of VSG mono-allelic expression are interesting and novel findings. However, I also agree that the manuscript is difficult to follow, even in its revised version. This is not an inevitable consequence of a manuscript containing a large amount of data as asserted in the rebuttal. It is a question of sentence structure and wording. I highly recommend that the authors go over the manuscript together with a native English speaker to improve clarity throughout.

We thank the reviewer for these comments. We have revised the manuscript extensively, paying particular attention to its clarity (new changes are marked in the submitted document).

In terms of the points made by reviewer 1, the revisions ultimately do not resolve which RNA sequence is bound by TbRAP1 and together with the non-sequence specific interactions with DNA and the unexpected binding to 170-no-VSG leave the puzzle of specificity in MAE regulation unanswered.

We would like to respectfully disagree with the reviewer on this point.

This manuscript's major conclusion is that *TbRAP1* contains an RRM domain that binds the active VSG RNA to sustain VSG MAE. This conclusion is supported by our NMR structure, *in vitro* binding studies, *in vivo* RNA CLIP, and mutational/functional analyses. Specifically, we used three independent *in vitro* binding assays - NMR titration, fluorescence polarization assay and EMSA - to show that the RRM and DB domains of *TbRAP1* cooperate for effective binding to the **consensus 16-mer in VSG 3'UTRs**. Structure-based mutational studies (including 2FQ, 2FA, 2FL, 5A and 2FA&5A) further confirmed that loss of RNA binding activity disrupted both aspects of VSG MAE: the active VSG is expressed at a much lower level and silent VSGs are derepressed. These results clearly explained (hence showed the mechanism of) how *TbRAP1* binds the active VSG RNA to regulate VSG MAE.

Regarding the reviewer's comment on specificity, we would like to point out that RRM domains are known to recognize a large number of short RNA sequences. Notable examples include hnRNPs and splicing factors, which recognize many RNA substrates. Therefore, it is most likely that the consensus 16-mer in VSG 3'UTRs is not the only sequence that *TbRAP1* recognizes. In fact, we observed that *TbRAP1* binds to the 170-no-VSG that does not contain any VSG2 3'UTR sequence, which also supports this prediction. We have emphasized this point in the revised manuscript (Page 15, lines 434 – 447).

The question remains of how many RNA substrates *TbRAP1* can recognize, and consequently, how many cellular processes can be affected by *TbRAP1*-RNA interactions. However, we would like to argue that identifying all possible RNA substrates for *TbRAP1* should not be a prerequisite for understanding the specific role of *TbRAP1* in VSG MAE. Our *in vitro* and *in vivo* data collectively shows that *TbRAP1* binds the active VSG RNA through its RRM domain, and this RNA binding activity is crucial for high-level transcription of the active VSG. Thus, our study uncovers novel insights into how *TbRAP1* plays a specific and essential role in VSG MAE.

It is not entirely clear to this reviewer why the authors did not expand on binding experiments with a systematic panel of short RNA substrates that would have defined the specific sequence motif recognized by the RRM.

We appreciate the reviewer's suggestion. Indeed, in the future, the natural progression of scientific studies would ideally determine the sequences of all *TbRAP1* RNA substrates.

However, as we stated above, such studies should not be a prerequisite for the current manuscript, nor would they contribute meaningfully to the findings reported here. This manuscript focuses on the fact that *TbRAP1* binds the active VSG RNA and this binding is essential for VSG MAE. Since we have clearly shown that *TbRAP1* can recognize the consensus 16-mer in VSG 3'UTRs, we have revealed at least one mechanism of how *TbRAP1* binds the active VSG RNA. Finding more RNA substrates of *TbRAP1* may or may not help explain how *TbRAP1* regulates VSG MAE because these sequences may not be related to VSG at all.

Additionally, such studies would also lie beyond the scope of the current manuscript. As we explained above, RRM domains are known to bind RNA promiscuously with low to no sequence preference. As a result, systematic screening to identify the specific RNA substrate(s) for an RRM domain is always very laborious and may lead to non-specific outcomes. For example, Fused in Sarcoma (FUS) is an RRM-containing RNA binding protein implicated in several neurodegenerative diseases. Despite extensive studies, the RNA sequence specificity of FUS and its RRM domain has not been fully characterized. Many papers have been published characterizing one or a few RNA sequences such as CG-, AUU- and GGU-containing motifs. Consequently, we believe our discovery of *TbRAP1*'s binding to the active VSG RNA through its RRM domain represents a comprehensive study suitable for publication.

Although the in vitro binding experiments are now presented in "a more scientific manner" by listing molar concentrations for substrate and protein, the method section makes it clear that the manuscript harbors a collection of binding assays carried out under different conditions. The reader is left wondering which condition applies to which assay. For example, whether EMSA assays are run through 0.8% agarose, 1.2% agarose, or 5% polyacrylamide can have profound effects on observed mobility shifts and apparent binding constants.

We are grateful to the reviewer for raising this point. We have now clearly marked which EMSA was run in what gel electrophoresis condition in the materials and methods section (Page 22, lines 625 – 628) and in figure legends for Figures 3, 7, and S3. We agree that different running conditions may affect the binding results. However, for the few EMSA experiments in which we actually used more than one running condition, same binding results were obtained, strongly suggesting that the differences between these running conditions do not affect the binding significantly. We believe that observing a positive binding result using more than one condition further validates that the binding activity is solid. In addition, we used the same running conditions whenever we compared binding affinities.

Furthermore, variable levels of radioactive material appear to be retained in the wells in all EMSA panels, but the gels are cropped in a manner that prevents further assessment of well-shifts. EMSA gels are not included in the source data raw data images.

We thank the reviewer for this comment. All of the EMSA images are already whole gel images. As a result, we did not include the EMSA images in the section for raw data images. Nevertheless, we have now added the EMSA raw images in the source data section.

In summary, the authors propose an interesting model in line with recent findings of how mutually exclusive DNA and RNA binding contributes to the regulation of transcription in a locus specific manner. Unfortunately, the manuscript lacks clarity, and the mechanism ultimately remains elusive.

We would like to respectfully disagree with the reviewer on this point. There are two points we would like to clarify further.

First of all, we thank the reviewer for pointing out that the manuscript needs language editing to improve its clarity (as discussed in reply to the first comment). We have revised the manuscript extensively, bearing in mind sentence structure and wording to communicate our findings more clearly (new changes are marked in the submitted document).

Secondly, we would like to point out that our *in vitro* and *in vivo* results present a clear mechanistic model of how the RRM-mediated RNA binding of *TbRAP1* regulates VSG MAE. Specifically, we have shown that *TbRAP1* has an RRM-mediated RNA binding activity *in vitro* and binds the active VSG RNA *in vivo* to sustain the full-level expression of the active VSG. Mechanistically, we have also shown that *TbRAP1* recognizes the consensus 16-mer in VSG 3'UTRs. In addition, we have shown that *TbRAP1*'s RNA and DNA binding activities are mutually exclusive and compete in a substrate concentration-dependent manner. As we have discussed above, although *TbRAP1* may recognize other RNA sequences, its binding to the consensus 16-mer in VSG 3'UTRs allows the abundant RNA of the active VSG to antagonize *TbRAP1*'s silencing effect and sustain its full-level transcription. Altogether, our study has identified *TbRAP1*'s RNA-binding activity as an unexpected and novel mechanism of VSG MAE. Our findings help to open up an uncharted area for a better understanding of antigenic variation in *T. brucei*.

Other comments:

Figure 1b, correct: CLIP/input

The normalization appears problematic as it artificially removes all variation among experiments for the IgG pulldown, thereby inflating significance of difference for RAP1 Ab.

We thank the reviewer for this comment. We have changed Fig. 1b to calculate the CLIP/Input value without normalizing it against the IgG result.

The (UUAGGG)₄ probe used to assess TERRA binding is problematic as it folds into highly stable intramolecular G-quadruplex structures making the RNA unavailable for sequence-specific binding. Commonly 3.5 repeats are used for this reason.

We thank the reviewer for raising this point. Indeed, under the buffer conditions of our NMR titration experiments, a subpopulation of the (UUAGGG)₄ oligo could form the G-quadruplex structure. To draw a clearer conclusion, we repeated the NMR titration analysis using a shorter probe of (UUAGGG)₂. Our data shows that *TbRAP1* RRM does not bind (UUAGGG)₂. We have updated the relevant sections in Fig. 2 and Supplementary Fig. S2 using the new data from (UUAGGG)₂.

It is surprising that 2FQ and 2FA mutants result in acute growth arrest, as expression of multiple VSGs would not be expected to cause this phenotype. The authors should comment on what is responsible for this acute phenotype.

We thank the reviewer for raising this point. VSG derepression is not the only phenotype observed in *TbRAP1* RRM point mutants. In fact, we observed an increased amount of DNA damage in 2FQ and 2FL mutants (Fig. 6). We have now performed western in the 2FA mutant, too, and detected an increased level of γ H2A signal after inducing Cre (Supplementary Fig 6h), indicating that 2FA is also defective in genome integrity. This is presumably the reason that these mutants have an acute growth defect. We have also added a short discussion about this in the manuscript (Page 13, lines 390 – 392).